# CLIP MODEL IS AN EFFICIENT ONLINE CONTINUAL LEARNER

## ABSTRACT

Online continual learning addresses the challenge of learning from continuous, non-stationary data streams. Existing online continual learning frameworks are classification-based and assume a pre-defined number of classes. In this study, we propose that vision-language models (VLMs) are more suitable candidates for online continual learning. Compared to traditional classification-based frameworks, VLM such as CLIP model is not limited by the maximum number of classes or constrained by rigid model architectures, enabling it to generalize across both known and emerging classes. However, we find that naively tuning the CLIP for online continual learning results in asymmetric image-text matching. This asymmetric matching will consistently poses negative suppression on the previously learned classes, leading to catestrophic forgetting. To address this issue, we propose a simple yet effective method, the symmetric image-text (SIT) tuning strategy, which mitigates the adverse impact of negative samples by excluding asymmetric text during online learning. Additionally, we introduce a more challenging online continual learning setting with blurred boundary, namely MiD-Blurry, which mixes multiple data distributions to simulate real-world scenarios. We conduct extensive experiments on several continual learning benchmarks as well as the MiD-Blurry setting, evaluating both inference-at-any-time performance and generalization to future data. Our results demonstrate that the SIT strategy effectively preserves memory stability while maintaining learning plasticity.

## 1 INTRODUCTION

Learning is the foundation for intelligent systems to adapt to the environment. Traditional supervised training paradigms have proven remarkably effective in closed or constrained environments where the data distribution remains relatively stable. However, in open real-world scenarios, the distribution of data may change over time. And due to constraints such as storage limitations and privacy concerns, it is often impractical to retain all data. An ideal artificial intelligence should continuously assimilate new knowledge from the dynamic environment, resembling human learning capabilities. Continual learning has emerged as a promising solution to address these challenges, but it faces the dilemma of the trade-off between learning plasticity and memory stability.

Offline continual learning permits the caching of data over extended periods and periodic model updates, assuming that the data distribution is stationary for a certain duration. The data stream can be segmented into a series of subsets (namely tasks or sessions) with clear boundaries. For instance, in Class-Incremental Learning (CIL) (Rebuffi et al., 2017), tasks have disjoint data label spaces. In contrast, online continual learning (Prabhu et al., 2020) focuses on a more practical scenario, where samples are accessible only during the current step and the model is required to inference at any time. In real-world scenarios, the assumption of data stability over short periods is often difficult to guarantee, leading to unclear boundaries between tasks potentially. Consequently, online continual learning is considered as a more challenging problem, as it demands models to dynamically adapt to non-stationary environments while retaining previously acquired knowledge.

Although some works (Koh et al., 2022; Wang et al., 2022; Moon et al., 2023) have attempted to address these challenges in online continual learning, these methods are all based on classification models, which are typically designed for closed-set scenarios that require pre-defining the maximum number of classes, thereby encountering various limitations. Such constraints hinder

their ability to adapt to the ever-evolving real-world data, where new classes and samples are continuously introduced without prior knowledge of their existence. Recent work (Thengane et al., 2022) has introduced to frozen pretrained Vision Language Models (VLMs), such as the Contrastive Language-Image Pretraining (CLIP) model (Radford et al., 2021), into class-incremental learning. Unlike conventional classifiers, the CLIP model performs classification by matching images to textual descriptions. This framework enables a more flexible learning process, which is particularly beneficial for online continual learning. However, we observe that although the frozen pre-trained CLIP model exhibits well generalization capabilities, its performance remains suboptimal in online continual learning scenarios. This is primarily due to the substantial distributional differences between the pre-training data and the downstream task data.

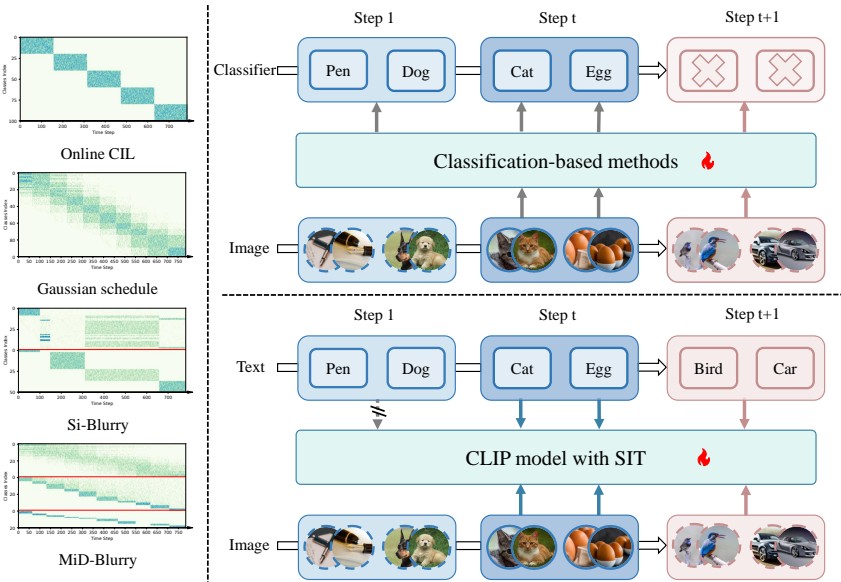

Figure 1: Framework of the proposed setting and method. Left: The online continual learning setting. Upper right: traditional classification-based online continual learning methods. Lower right: VLM-based online continual learning methods utilizing the SIT strategy.

One straightforward solution is the Parameter-Efficient Fine-Tune (PEFT) approach to enhance the plasticity of the CLIP model. PEFT allows the model to quickly adapt to downstream tasks by fine-tuning only low-parameter adapters. However, directly fine-tuning the CLIP model, even when combined with classic online continual learning method, can still lead to serious catastrophic forgetting. We observe that the asymmetry between image and text during fine-tuning, specifically matching all previously seen classes with images at the current step, is the main reason. Through gradient analysis, we find that the asymmetry causes an imbalance in the gradients of positive and negative samples, leading the model to bias toward predicting all classes as new. To address this, we propose a simple yet effective method, the symmetric image-text (SIT) strategy, which effectively prevents erroneous model updates by removing asymmetric negative texts in online learning. Without bells and whistles, the experiments conducted across multiple continual learning datasets and in various online continual learning settings demonstrate the effectiveness of our proposed SIT strategy. It not only enhances the model's plasticity, allowing effective adaptation to new data, but also maintains memory stability. Additionally, feature visualization aids in intuitively understanding the role of SIT, further proving its effectiveness in improving model performance and stability in online continual learning scenarios.

Our contributions can be summarized as follows:

- We explore the application of vision-language models in the context of online continual learning. By leveraging an open-vocabulary approach, the CLIP model avoids the limitations imposed by traditional model architectures, such as predefined the maximum number of classes, enabling real *endless learning*.

- Through theoretical analysis and experimental validation, we identify that the asymmetry between image and text features in the CLIP model is a key contributor to catastrophic forgetting in online continual learning. To address this, we propose a simple yet effective online continual learning method, namely Symmetric Image-Text (SIT) strategy, which effectively maintaining model plasticity while maintains memory stability.

- We introduce a more challenging online continual learning setting, MiD-Blurry, which mixes multiple training data distributions to better simulate real-world scenarios. We conduct extensive experiments on several online continual learning benchmarks including Si-Blurry as well as our MiD-Blurry setting. The results demonstrate that our SIT strategy outperforms both classification-based approaches and other CLIP fine-tuning methods that incorporate continual learning techniques.

## 2 RELATED WORK

**Classification-based Continual Learning.** To alleviate catastrophic forgetting, various continual learning methods have been proposed. Existing continual methods can be categorized into three types (Zhou et al., 2023): data-based (Rebuffi et al., 2017; Bang et al., 2021), algorithm-based (Kirkpatrick et al., 2017; Li & Hoiem, 2017; Hou et al., 2019; Zhu et al., 2021) and model-based methods. Recently, pioneering works (Zhou et al., 2022b; Wang et al., 2022) utilize pre-trained models and introduce prompt-tuning to balance the stability and plasticity of the model, demonstrating the superiority of pre-trained models in continual learning (Moon et al., 2023) employs instance-wise logit masking and contrastive visual prompt tuning loss to reinforce the retention of previously learned knowledge while adapting to new tasks. However, these continual learning methods are all based on a classification model, which requires a predefined maximum number of classes to determine the dimension of classifiers. As a result, they are considered closed-ended continual learning methods, making them unsuitable for handling the open nature of real-world scenarios.

**VLM-based Continual Learning.** Thengane et al. (2022); Zheng et al. (2023); Yu et al. (2024) have attempted to integrate the pre-trained CLIP model into continual learning. However, Continual-CLIP (Thengane et al., 2022) sacrifices the model's plasticity entirely, which disqualifies it from being considered a true continual learning method. Moreover, experiments (Zheng et al., 2023; Yu et al., 2024) demonstrate that even the CLIP model, pre-trained on large-scale datasets, struggles to perform well on downstream tasks with data distributions different from those in training. As a result, these works focus more on how continual learning impacts the zero-shot performance of CLIP model. However, these works are limited to offline class-incremental learning setting. It is essential to explore the application of VLM-based methods in more challenging online continual learning scenarios. Adapting to gradually changing data distributions is the central challenge of online continual learning, as well as the limitations imposed by predefined maximum class numbers, which can be bypassed by utilizing the image-text matching framework. Therefore, we propose that the CLIP model is an efficient online continual learner.

**Open World Recognition.** Traditional image classification models operate under the closed-world assumption, where all test classes are seen during training. To address the challenge of open-world recognition, several representation learning methods (Zhao et al., 2021; Kim et al., 2025) have been proposed. These methods adjust the distances between feature representations based on semantic similarity to obtain generic and discriminative features, enabling the distinction of unseen classes. Recent advancements in vision-language models have further facilitated the classification of unseen classes by matching images with textual descriptions of classes. The CLIP model, as a continual learner, inherently supports open-vocabulary image classification. Its zero-shot performance can be evaluated to analyze how continual learning impacts its pretrained knowledge retention. In this field, the most closely related works to ours are TreeProbe (Zhu et al., 2023), MoEAdapter (Yu et al., 2024), and AnytimeCL (Zhu et al., 2025). These methods adopt the concept of weight ensembling, balancing stability and plasticity by freezing the pretrained CLIP model and introducing new learnable branches. However, unlike our method, MoEAdapter and AnytimeCL simplify the problem to task-incremental learning by fitting the distribution of seen classes, making them less effective in handling shifts in data distribution. Moreover, both TreeProbe and AnytimeCL heavily rely on a large number of exemplars, which limits their scalability.

**Parameter-Efficient Tuning in CLIP model.** Trained with abundant available data from the web, the vision-language model CLIP (Radford et al., 2021) demonstrate great advantage in a wide variety of tasks including few-shot and zero-shot visual recognition. However, how to efficiently adapt it to downstream tasks still remains a challenge. To solve this problem, several parameter-efficient tuning methods have been proposed, roughly categorized into prompt-tuning Lester et al. (2021) and prefix-tuning (Li & Liang, 2021), LoRA (Hu et al., 2022), Adapter (Houlsby et al., 2019). Inspired by prompt learning in NLP, many works tune CLIP through the learnable prompt (Zhou et al., 2022a) applies prompt learning-based approach to CLIP for the first time and shows exceptional performance in downstream transfer learning. Khattak et al. (2023) improves the alignment between two modalities by projecting textual prompt into visual prompt and embedding them into corresponding encoders. Wang et al. (2023) enhances generalizability and mitigates forgetting by using orthogonal prompts as attributes.

## 3 METHODOLOGY

### 3.1 PROBLEM FORMULATION

Continual learning aims to train a unified model $\mathcal{F}_\theta : \mathcal{X} \to \mathcal{Y}$ parameterized by $\theta$ that makes good predictions for all seen classes. In classic class-incremental learning setting, given a sequence of tasks $\mathcal{T} = \{\mathcal{T}_1, \mathcal{T}_2, ..., \mathcal{T}_T\}$, the training set of the $\tau^{th}$ task $\mathcal{T}_\tau$ is $\mathcal{D}_\tau = (\boldsymbol{x}_i^\tau, y_i^\tau)_{i=1}^N$, where $\boldsymbol{x}_i^\tau \in \mathcal{X}$ and $y_i^\tau \in \mathcal{Y}$ denote the input sample and its corresponding label respectively. We define the output space for all observed class labels $\mathcal{Y}^{(\tau)} \subset \mathcal{Y}^{(\tau+1)}$. In offline continual learning, a task $\mathcal{T}$ represents a specific time period during which the data distribution remains stationary.

In contrast, online continual learning allows access only to the current batch of training data $\mathcal{B}_t = (\boldsymbol{x}_i^t, y_i^t)_{i=1}^N$ at each time step $t^{th}$. Noted that the task $\mathcal{T}$ to represent a sudden change in data distribution, but the learner is unaware of the task alteration during training. Figure 1 illustrates several online continual learning settings. In the first subplot of the left section of Figure 1, online CIL directly applies the class-incremental learning setting to an online scenario. The label $\mathcal{Y}^{(t)}$ between any two tasks are disjoint, i.e. $\mathcal{Y}^{(t)} \cap \mathcal{Y}^{(t')} = \emptyset$. We denote $\tau$ as the time step at which task $\mathcal{T}_\tau$ begins. The Gaussian schedule (Shanahan et al., 2021) builds on this by assuming that samples should follow a Gaussian distribution. Specifically, for a sample $(\boldsymbol{x}_i^t, y_i^t)$, its distribution is defined as $P((\boldsymbol{x}_i^t, y_i^t) \in \mathcal{T}_\tau) = \frac{1}{\sigma\sqrt{2\pi}} e^{-\frac{(t-\mu)^2}{2\sigma^2}}$. The i-Blurry (Koh et al., 2022) assumes that some classes are uniformly distributed across all time steps, aside from disjoint classes. The Si-Blurry (Moon et al., 2023), as shown in Figure 1, further posits that the number of classes appearing within a given time period should also be random. Additionally we introduce a more challenging benchmark that better simulates the distribution of non-stationary data streams in the real world, namely MiD-Blurry, as shown in the last subplot of the left section of Figure 1, where the training data follows a mixed distribution. In addition to disjoint classes and Gaussian classes, we also design a decay classes whose distribution follows $P((\boldsymbol{x}_i^t, y_i^t) \in \mathcal{T}_\tau | t \geq \tau) = 1/t^\alpha$ to simulate phenomena that things suddenly appear and gradually fade away. The tasks $t$ where the means $\mu$ of the Gaussian classes and the decay classes first appear are uniformly distributed, and the number of classes in each task is random. In Section A.1, we visualize the data distributions for the aforementioned online continual learning settings.

### 3.2 CLIP MODEL AS AN ONLINE CONTINUAL LEARNER

The CLIP model represents a significant advancement in multi-modal machine learning. It is designed to map images and texts into a shared feature space, with maximizing the similarity between feature vectors of image-text pairs. The CLIP model lies in its ability to learn rich joint representations, and it can capture the nuanced relationships between visual content and textual descriptions, which is crucial for tasks that involve understanding and generating language conditioned on images. As shown in Figure 2, considering a $K$-class image classification problem, CLIP maps an unidentified image $\boldsymbol{x} \in \mathcal{X}$ to its corresponding feature vector through the image encoder $\boldsymbol{v} = \mathbf{E}_{\texttt{visual}}(\mathbf{x})$. Class label $y \in \mathcal{Y}$ is prepended by a hand-crafted prompt template $\mathbf{p} \to$ *a photo of a {class}.* to form a class-specific text input $\mathbf{y} = \{\mathbf{p}; y\}$, which is then encoded into a text feature vector $\boldsymbol{t}$ by the

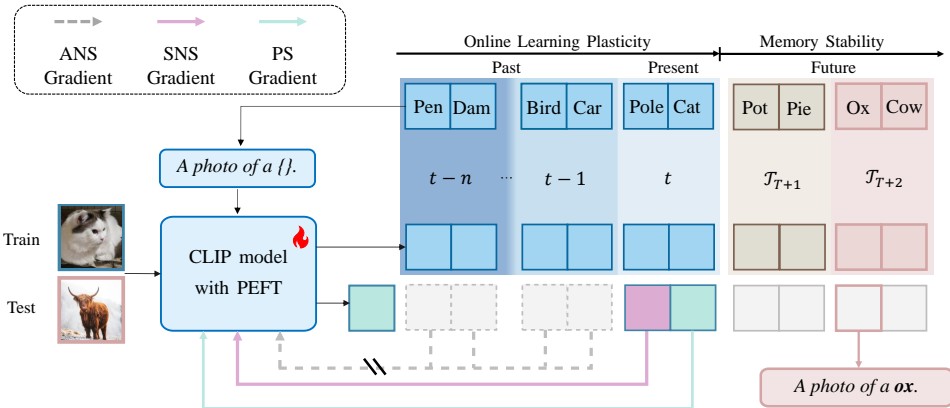

Figure 2: The CLIP model serves as an online continual learner. By achieving classification through matching images with texts of class names, the CLIP model avoids structural limitations and enables *endless learning*. Our proposed SIT strategy mitigates catastrophic forgetting by correcting the model's update direction during learning by removing asymmetric negative samples (ANS).

text encoder $t = \mathbf{E}_{\text{text}}(\mathbf{y})$. The prediction probability can be denoted by

$$p(y_i|\boldsymbol{x}) = \frac{\exp(\text{sim}(\boldsymbol{t}_i \cdot \boldsymbol{v}))}{\sum_{i=1}^{K} \exp(\text{sim}(\boldsymbol{t}_i \cdot \boldsymbol{v}))}, \tag{1}$$

where $\text{sim}(\cdot)$ denotes the cosine similarity.

Traditional classification-based continual methods are constrained by a predefined set of classes, which necessitates model retraining or adjusting when novel classes are introduced in continual learning. In contrast, the design of CLIP model overcomes these limitations by employing a open-vocabulary manner. This approach allows CLIP to dynamically adapt to new classes without altering the model architecture. The incorporation of textual descriptions as classifiers provides a flexible and scalable solution for continual learning. This capability is particularly advantageous in environments where the set of classes is continuously expanding. Moreover, the generalizability afforded by pre-training on large-scale datasets, along with the rich semantic information embedded in class texts, can aid CLIP in generalizing to downstream tasks. This characteristic allows us to evaluate the model's adaptability to future data in a zero-shot manner, in addition to handle previously seen classes. Therefore, we believe that CLIP model represents a more efficient online continual learner.

### 3.3 Symmetric Image-Text tuning strategy

Despite the CLIP model's pre-training on large-scale datasets, its performance on online continual learning scenarios with significantly different distributions remains suboptimal. Adapting the model to gradually changing distributions is a common scenario for online continual learning. Thus, parameter efficient fine-tuning (PEFT) for the CLIP model is essential. After adding an adapter or prefix to the CLIP model, it is typical to tune with the InfoNCE loss:

$$\mathcal{L}_{\text{infoNCE}} = -\sum_{\boldsymbol{v}_i \in \boldsymbol{V_b}} \log \frac{\exp(\text{sim}(\boldsymbol{v}_i, \boldsymbol{t}_+)/\tau)}{\sum_{\boldsymbol{t}_j \in \boldsymbol{T}} \exp(\text{sim}(\boldsymbol{v}_i, \boldsymbol{t}_j)/\tau)}, \tag{2}$$

where $\boldsymbol{v}_i$ and $\boldsymbol{t}_+$ are the positive sample pairs, $\boldsymbol{V_b}$ is the image feature vectors of the current batch, and $\boldsymbol{T}$ is the text feature vectors of all seen classes.

However, experiments indicate that directly using this loss for fine-tuning leads to severe catastrophic forgetting. To analyze this question, we compute the gradient norms during the training process, which indicates the strength of sample contributions to the updates of parameters associated with each class at each step. Specifically, considering that in online continual learning scenarios, the model can only access to the current batch of images at each step, while all seen classes are known, we can categorize text features into symmetric text features and asymmetric text features. For an image feature $\boldsymbol{v}_i$, we denote $\boldsymbol{t}_+$ as the positive sample (PS), $\boldsymbol{t} \in \boldsymbol{T_s}$ as a symmetric negative sample

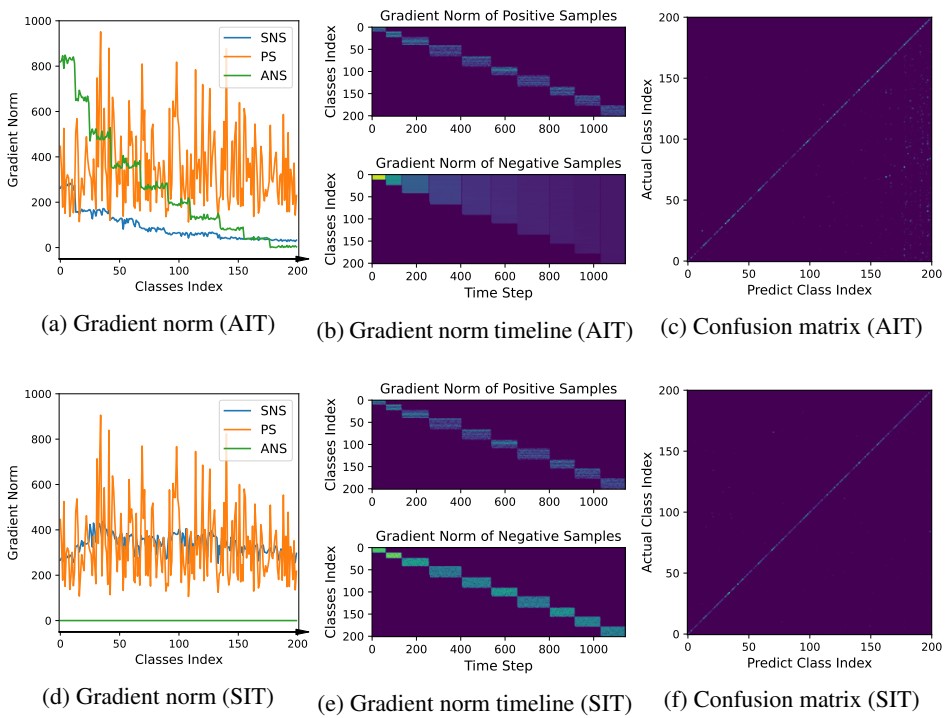

Figure 3: Confusion Matrix and Gradient Analysis of AIT and SIT. Gradients are computed for each class, with class indices sorted by their first appearance time. The norm of the gradient reflects the strength of the parameters updates associated with a particular class. An imbalance in the gradients of positive and negative samples can introduce bias. The gradient norm timeline is employed to observe the updates of parameters across both time and class simultaneously. In AIT, even in the absence of corresponding images, the ANS still leads to model updates, which can result in forgetting. To simplify the problem, we adopted the online class-incremental learning setting in this experiment, where all classes are treated as disjoint. The figures are generated using CIFAR-100, under the same settings described in Section 4.2 (seed=2024). Best viewed in color.

(SNS), and $t \in T_a$ as an asymmetric negative sample (ANS). Here, $T_s$ represents the text features corresponding to the classes of the images in the current batch, while $T_a$ represents other classes.

From Figure 3a, two issues can be observed. Firstly, the gradient of positive samples fluctuates around a certain value. For earlier-seen classes, the gradient when they act as negative samples is higher than when they act as positive samples, while for later-seen classes, the opposite is true. This indicates a bias towards newer classes during training, as the model tends to predict all classes as newer classes—a typical sign of catastrophic forgetting, as shown in Figure 3c. Secondly, the gradient contribution of most classes when they act as ANS is significantly higher than when they act as SNS or PS, suggesting that the issue likely lies in ANS. To address this, we attempted to isolate the ANS-related components from the loss function:

$$\mathcal{L}_{\text{infoNCE}} = \mathcal{L}_{\text{S}} + \mathcal{L}_{\text{A}} = -\sum_{\boldsymbol{v_i} \in \boldsymbol{V_b}} \log \frac{\exp\left(\text{sim}(\boldsymbol{v_i}, \boldsymbol{t_i^+})/\tau\right)}{Z_S} - \sum_{\boldsymbol{v_i} \in \boldsymbol{V_b}} \log \frac{Z_S}{Z_S + Z_A}, \tag{3}$$

where

$$Z_S = \sum_{\boldsymbol{t_j} \in \boldsymbol{T_{S+}}} \exp\left(\text{sim}(\boldsymbol{v_i}, \boldsymbol{t_j})/\tau\right), Z_A = \sum_{\boldsymbol{t_k} \in \boldsymbol{T_A}} \exp\left(\text{sim}(\boldsymbol{v_i}, \boldsymbol{t_k})/\tau\right), \tag{4}$$

and $\boldsymbol{T_{S+}} = \boldsymbol{T_S} \cup \{\boldsymbol{t_+}\}$.

Clearly, $\mathcal{L}_S$ is a standard infoNCE loss that brings the $\boldsymbol{v_i}$ closer to PS while pushing it away from SNS. $\mathcal{L}_A$ determines the relative relationship between the $\boldsymbol{v_i}$ and ANS. Optimizing $\mathcal{L}_A$ indirectly pushes the distance between the $\boldsymbol{v_i}$ and ANS, which can introduce bias.

To counteract this issue, we propose a simple yet effective method called the Symmetric Image-Text (SIT) strategy. As shown in Figure 2, SIT strategy discards ANS, which can lead to bias in online continual learning, restoring the symmetry between image and text features during training. Specifically, we reformulate the infoNCE loss function:

$$\mathcal{L} = -\sum_{\boldsymbol{v_i} \in \boldsymbol{V_b}} \log \frac{\exp(\text{sim}(\boldsymbol{v_i}, \boldsymbol{t_+})/\tau)}{\sum_{\boldsymbol{t_j} \in \boldsymbol{T_{s+}}} \exp(\text{sim}(\boldsymbol{v_i}, \boldsymbol{t_j})/\tau)}. \tag{5}$$

By doing so, we effectively mitigate catastrophic forgetting, allowing the model to maintain its zero-shot learning capabilities while adapting to new classes in an online continual learning context. As shown in Figure 3d, the gradient norms for SNS are generally balanced with PN. Figures 3b and Figures 3e demonstrate the temporal evolution of the gradients, allowing us to observe the effects of gradient norms on the model across both class and temporal dimensions. Each point reflects the influence of samples from a specific class on model parameters at a given step. For SIT, Figures 3e illustrates that a class impacts model parameters only during its appearance. In contrast, for AIT, i.e. the vanilla method, classes continue to affect model parameters even after their initial appearance, leading to forgetting when no positive samples are present to reinforce learning. This indicates that SIT is more effective in preserving previously learned knowledge.

## 4 EXPERIMENTS AND RESULTS

### 4.1 EXPERIMENTAL DETAILS

**Datasets and settings.** We conducted extensive experiments across a variety of datasets to evaluate the performance of our proposed SIT strategy in continual learning, as well as its ability to maintain zero-shot transfer capability. In the experimental setting, in addition to the proposed MiD-Blurry, we conducted experiments on Si-Blurry and class-incremental learning. Unless otherwise specified, for MiD-Blurry, the number of tasks is set to 10, with 30% of the classes designated as Gaussian classes ($\sigma = 0.15$), 30% as decay classes ($\alpha = 1.5$), while the remaining classes are disjoint classes. For Si-Blurry, the disjoint class ratio $N = 50$, blurry level $M = 10$ and task number $T = 5$. For general datasets, we selected CIFAR-100 (Krizhevsky et al., 2009), TinyImageNet (Le & Yang, 2015), and Caltech101 (Fei-Fei et al., 2004). For fine-grained downstream tasks, we chose Flowers102 (Nilsback & Zisserman, 2008), OxfordPets (Parkhi et al., 2012), Food101 (Bossard et al., 2014), the scene understanding dataset SUN397 (Xiao et al., 2010), the bird image dataset CUB200 (Wah), StanfordCars (Krause et al., 2013), FGVCAircraft(Maji et al., 2013), and the satellite image dataset EuroSAT (Helber et al., 2019). Additionally, to evaluate adaptability to different domain data, we conducted experiments on ImageNet-R (Hendrycks et al., 2021).

**Implementation Details.** We conducted experiments on a pre-trained ViT-B/16 CLIP model (Radford et al., 2021), where $d_l = 512$, $d_v = 768$ and $d_{vl} = 512$. The prompt template utilizes *a photo of a {class}*.. Unless stated otherwise, the PEFT method we employed is LoRA (Hu et al., 2022), with a rank of 4, which is integrated into every transformer layer of the image and text encoders in the CLIP model. In the online continual learning, the batch size for each step is set to 64. We update each batch three times using the Adam optimizer, with a learning rate of $5e - 4$. No techniques such as experience replay or knowledge distillation are employed. For the implementation of MaPLe (Khattak et al., 2023), we set the prompt depth $J$ to 9 and standardized the lengths of both the language and vision prompts to 2. MaPLe is optimized using the SGD optimizer with a learning rate of $0.0001$, also over 3 iterations per batch.

**Evaluation Metrics.** We record the top-1 accuracy $\mathcal{A}_t$ of the model on the test set after finishing training at step $t$ and present it as a curve, where the test set contains all classes the model has ever seen. We denote the accuracy at the end of the last task $\mathcal{A}_{\text{last}}$ as a metric for overall accuracy. To evaluate the online learning ability of the model, we also use $\mathcal{A}_{\text{auc}}$ (Koh et al., 2022) to measure the performance of anytime inference, which assumes that inference queries can be made anytime during training. $\mathcal{A}_{\text{auc}} = \sum_{i=1}^{k} f_A(i \cdot \delta_n) \cdot \delta_n$, where $\delta_n$ is the number of seen samples during the

evaluation and $f_A(\cdot)$ is the accuracy curve. And for class-incremental learning setting, we use the average of the test accuracy across all tasks $\mathcal{A}_{\text{avg}} = \frac{1}{T} \sum_{t=0}^{T-1} \mathcal{A}_t$ to evaluate the overall performance.

## 4.2 ONLINE LEARNING PLASTICITY EVALUATION

This experiment aims to evaluate the plasticity of models across different online continual learning methods. Table 1 shows the performance of online continual learning on the MiD-Blurry setting. In the comparative methods, DualPrompt (Wang et al., 2022) and MVP (Moon et al., 2023) are classification-based methods, fine-tuning the pre-trained ViT-B/16 model (Dosovitskiy, 2020) through prompt learning. The remaining methods are VLM-based, where Continual-CLIP (Thengane et al., 2022) does not perform any fine-tuning on the model. AIT-CLIP, SIT-CLIP, ER (Rolnick et al., 2019), and LwF (Li & Hoiem, 2017) utilize LoRA for fine-tuning, with ER (Rolnick et al., 2019) having a memory size of 1,000 samples. For the PEFT method MaPLe (Khattak et al., 2023), we also set the memory to 1,000 samples. MVP-CLIP replaces the backbone network and classifier of MVP with the image and text encoders of the CLIP model. For MoEAdapter (Yu et al., 2024), the settings from its original paper are applied. The results demonstrate that VLM-based methods generally outperform classification-based approaches in the online continual learning setting, particularly under the MiD-Blurry conditions. For instance, while the DualPrompt method yields an accuracy of 73.63% on CIFAR-100, the Continual-CLIP method achieves a competitive 72.66%, highlighting the efficacy of VLM architectures. Notably, the comparison with the Continual-CLIP indicates that online continual learning significantly enhances model performance, as evidenced by the improvements across various datasets. Our proposed method, SIT-CLIP, distinguishes itself by achieving the highest accuracy metrics, such as 84.34% on CIFAR-100, while employing a minimal number of trainable parameters (0.37 M). This efficiency is particularly remarkable given that SIT-CLIP does not rely on techniques such as experience replay or knowledge distillation, which are often employed by other methods. Overall, SIT-CLIP demonstrates superior performance with fewer resources, reinforcing its potential for effective online continual learning. Furthermore, we explored different data distributions in the MiD-Blurry setting, and the corresponding results are included in Section A.3.

Table 1: Performance comparison of online continual learning on MiD-Blurry setting. #P represents the number of trainable parameters, and #TP represents the total number of parameters. This experiment aims to evaluate the any time inference performance during online continual learning, as well as its final classification accuracy.

| Method | #P | #TP | CIFAR-100 | | TinyImageNet | | ImageNet-R | |
|---|---|---|---|---|---|---|---|---|
| | | | $\mathcal{A}_{\text{auc}}$ | $\mathcal{A}_{\text{last}}$ | $\mathcal{A}_{\text{auc}}$ | $\mathcal{A}_{\text{last}}$ | $\mathcal{A}_{\text{auc}}$ | $\mathcal{A}_{\text{last}}$ |
| DualPrompt | 0.55 M | 86.35 M | 73.63±1.10 | 58.31±2.34 | 65.92±1.09 | 50.80±1.42 | 35.42±0.39 | 29.56±1.02 |
| MVP | 0.55 M | 86.35 M | 75.70±0.93 | 67.70±1.46 | 70.66±1.02 | 63.61±0.77 | 34.02±0.32 | 30.84±1.31 |
| Continual-CLIP | 0.00 M | 149.62 M | 72.66±0.94 | 66.27±0.00 | 70.37±0.18 | 64.46±0.00 | 75.76±0.29 | 71.06±0.35 |
| AIT-CLIP | 0.37 M | 149.99 M | 74.52±0.45 | 59.52±1.17 | 70.56±1.26 | 56.89±0.66 | 76.99±0.68 | 65.21±1.32 |
| ER | 0.37 M | 149.99 M | 82.77±0.37 | 78.00±1.23 | 77.15±0.34 | 68.89±0.66 | **81.76±0.19** | 76.05±0.15 |
| LwF | 0.37 M | 149.99 M | 72.60±0.65 | 57.60±2.87 | 68.30±1.29 | 53.58±1.08 | 77.04±0.58 | 66.24±1.24 |
| MaPLe | 1.19 M | 150.81 M | 73.25±0.68 | 66.18±1.64 | 69.67±0.62 | 63.19±0.61 | 78.31±0.38 | 74.15±0.41 |
| MVP-CLIP | 0.48 M | 150.10 M | 60.44±0.93 | 45.87±0.31 | 56.65±1.34 | 39.31±2.08 | 66.53±1.14 | 57.79±1.18 |
| MoEAdapter | 4.03 M | 153.65 M | 81.88±0.17 | 77.39±0.27 | 78.17±0.68 | 74.05±0.64 | 81.30±0.12 | 76.88±0.16 |
| **SIT-CLIP (Ours)** | 0.37 M | 149.99 M | **84.34±0.56** | **79.47±0.52** | **79.88±0.68** | **75.50±0.20** | 81.65±0.18 | **77.53±0.65** |

In our detailed experiments conducted on the Si-Blurry setting, we further validated the effectiveness of our proposed SIT strategy, with results presented in Section A.4 due to page limitations. Additionally, we compared SIT strategy with state-of-the-art CIL methods in Section A.5, which also demonstrated a significant improvement in continual learning performance.

## 4.3 MEMORY STABILITY EVALUATION

In this section, we evaluate the zero-shot performance of the VLM-based methods. The experimental setting mirrors 4.2, focusing on the performance of fine-grained downstream datasets in online continual learning. The results are summarized in Table 2. It is noteworthy that most methods demonstrate commendable memory stability, with minimal degradation in zero-shot performance

compared to the non-finetuned baseline, Continual-CLIP. Specifically, the model's ability to retain knowledge from various datasets is evidenced by superior performance on certain datasets relative to Continual-CLIP. This indicates that the model effectively assimilates generalized knowledge from these fine-grained datasets, thereby enhancing its capabilities across other downstream tasks. Among the evaluated methods, Ours SIT-CLIP achieves the highest average accuracy of 70.24%, outperforming several established approaches. Notably, the strong performance of the ER method can be attributed to its smaller training dataset coupled with a larger amount of replay data, which enhances its memory stability. In contrast, MoEAdapter benefits from a greater number of training parameters and the Mixture of Experts (MoE) structure, contributing to its robust performance. The results underscore the potential of leveraging fine-grained datasets to bolster model performance, highlighting the effectiveness of our proposed approach.

Table 2: Comparison of zero-shot performance after online continual learning in CUB200, Stanford-Cars, FGVCAircraft with the MiD-Blurry setting. The best results are in bold, and the second-best results are underlined. This experiment aims to evaluate the memory stability of the model after online continual learning on multiple fine-grained datasets.

| Method | CUB200,StanfordCars,FGVCAircraft | | Targets | | | | | | |
|---|---|---|---|---|---|---|---|---|---|
| | $\mathcal{A}_{auc}$ | $\mathcal{A}_{last}$ | Flowers102 | OxfordPet | EuroSAT | Food101 | SUN397 | Caltech101 | Average |
| *Continual-CLIP* | *57.46±0.39* | *48.72±0.00* | *65.86* | *85.31* | *40.24* | *86.34* | *61.54* | *87.95* | *70.50* |
| AIT-CLIP | 59.32±0.07 | 43.69±1.38 | 64.25±1.60 | 84.31±0.48 | 26.88±4.95 | 83.39±0.58 | 59.79±0.62 | 84.83±4.95 | 66.67±1.70 |
| ER | 65.37±0.39 | 55.48±0.35 | 64.86±1.63 | 86.55±0.34 | 31.80±3.78 | 83.35±1.11 | 61.63±0.84 | 89.00±1.75 | 68.49±0.84 |
| LwF | 59.14±0.07 | 41.56±0.89 | 63.49±0.07 | 85.53±0.48 | 33.64±1.44 | 81.86±1.73 | 60.98±0.46 | 87.01±0.56 | 67.93±0.06 |
| MaPLe | 57.00±0.79 | 49.56±0.64 | 66.11±0.78 | 76.99±8.62 | **40.31±3.27** | **86.67±0.70** | 59.42±3.27 | 89.42±2.73 | 68.97±2.81 |
| MVP-CLIP | 47.45±0.31 | 35.27±0.82 | 63.77±0.84 | 83.06±0.71 | 28.70±4.48 | 78.93±0.59 | 58.54±0.23 | 87.98±0.93 | 65.87±0.89 |
| MoEAdapter | 64.93±0.84 | 56.96±0.45 | **69.05±0.13** | **87.32±0.46** | 29.56±1.16 | 84.09±0.76 | 61.57±0.38 | 91.02±0.97 | 69.62±0.09 |
| SIT-CLIP | **65.86±0.24** | **57.05±0.86** | 67.59±0.94 | 86.68±0.73 | 38.32±2.28 | 79.49±6.41 | **63.66±0.41** | **91.15±2.13** | **70.24±1.49** |

## 4.4 ABLATION STUDY

### 4.4.1 ANALYSIS OF SYMMETRIC IMAGE-TEXT TUNING STRATEGY

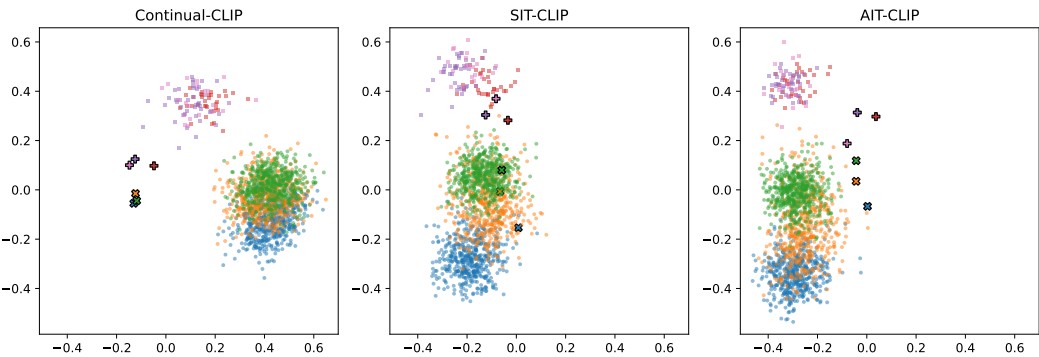

Figure 4: Visual comparison of AIT and SIT model features, where $\times$ and $\bullet$ represent text and image features from online continual learning datasets, and $+$ and $\square$ represent features from zero-shot datasets. Features are concatenated and reduced via PCA, with a consistent coordinate across the figure. The figures are generated using CIFAR-100, under the same settings described in Section 4.2 (seed=2024). Six classes are randomly selected from CIFAR-100 (CL) and Flowers102 (Zero-shot). Features are extracted offline after continual learning. Best viewed in color.

As discussed in Section 3.3, the asymmetry between image and text can lead to a degradation in the performance of online continual learning. By decomposing the infoNCE loss, we observed that $\mathcal{L}_A$ pushes image features away from text features, as visualized in the feature space. As shown in Figure 4, the experiments followed the settings in Table 1, and after training, we concatenated the features and applied PCA, allowing for direct comparison in the same coordinate. For Continual-CLIP, the pre-trained CLIP model shows that features from the same dataset are relatively dense, with a considerable distance between image and text features. In contrast, SIT disperses the image features through learning, achieving better alignment between image and text features, while also bringing the distances closer in the zero-shot dataset. The relative positioning of the features form

zero-shot dataset remains similar to the pre-trained model, indicating that SIT maintains memory stability and facilitates knowledge transfer to the future. AIT exhibits a distribution of image features similar to SIT, but the distance between image and text features increases, reflecting the respective functions of $\mathcal{L}_S$ and $\mathcal{L}_A$.

### 4.4.2 EFFECTS OF FINE-TUNING THE IMAGE ENCODER VERSUS THE TEXT ENCODER

Table 3: Comparative analysis of fine-tuning only the image encoder versus the text encoder. This experiment aims to investigate the respective roles of image and text encoders in online continual learning.

| Method | #P | #TP | $\mathcal{A}_{\text{auc}}$ | $\mathcal{A}_{\text{last}}$ | Zero-shot |
|---|---|---|---|---|---|
| *Continual-CLIP* | *0.00 M* | *149.62 M* | *72.66±0.94* | *66.27±0.00* | *62.06±0.00* |
| SIT-CLIP (Image only) | 0.22 M | 149.84 M | 81.97±1.04 | 75.85±0.71 | 59.89±0.68 |
| SIT-CLIP (Text only) | 0.15 M | 149.77 M | 77.43±0.62 | 71.40±0.84 | 57.99±1.45 |
| SIT-CLIP | 0.37 M | 149.99 M | 84.34±0.56 | 79.47±0.52 | 60.06±0.51 |

We compared the effects of fine-tuning only the text encoder versus the image encoder. The results presented in Table 3 demonstrate that fine-tuning either encoder significantly enhances the performance of online continual learning across all datasets, with the image branch yielding superior results. As illustrated in Figure 5, fine-tuning the image branch enhances the distinguishability of image features and aligns them more closely with text features, thereby effectively improving the model's performance. In contrast, fine-tuning only the text branch merely enhances the distinguishability of text features, resulting in performance that is inferior to that achieved by fine-tuning the image branch.

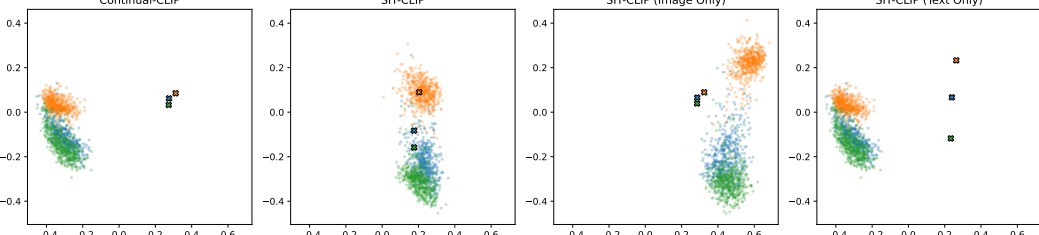

Figure 5: Visualization of features after fine-tuning different encoders, where $\times$ and $\bullet$ represent text and image features. Features are concatenated and reduced via PCA, with a consistent coordinate across the figure. The figures are generated using CIFAR-100, under the same settings described in Section 4.2 (seed=2024). Three classes are randomly selected from CIFAR-100. Best viewed in color.

## 5 CONCLUSION

Online continual learning involves the capability of models to learn from data streams while allowing for the evaluation of model performance at any moment. In this paper, we propose that vision-language models, such as the CLIP model, are more suitable candidates for online continual learning compared to traditional classification-based methods. Through analyzing the gradients and loss functions, we identified that the asymmetry between text and image in online continual learning is a significant cause of catastrophic forgetting. To mitigate this issue, we introduced a simple yet effective method known as the Symmetric Image-Text strategy, which removes the asymmetry of text in online continual learning. Furthermore, we present a more challenging online continual learning setting, MiD-Blurry, which better simulates real-world scenarios by mixing various data distributions. We conducted extensive experiments across multiple continual learning datasets, including Si-Blurry and MiD-Blurry settings. The results indicate that the SIT strategy effectively preserves memory stability while maintaining learning plasticity.

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

# A APPENDIX

## A.1 VISUALIZATION OF TRAINING DATA DISTRIBUTIONS FOR ONLINE CONTINUAL LEARNING

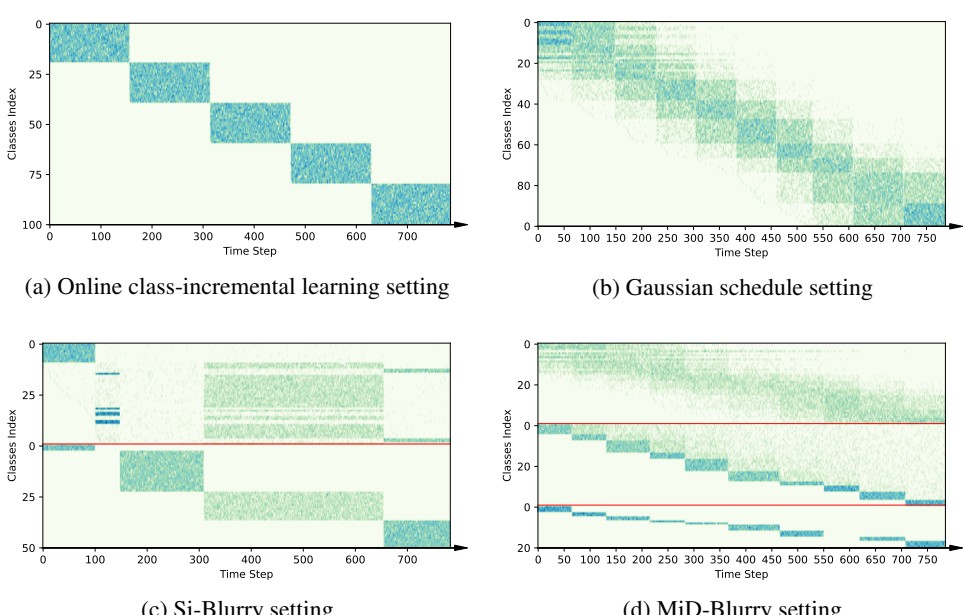

(a) Online class-incremental learning setting

(b) Gaussian schedule setting

(c) Si-Blurry setting

(d) MiD-Blurry setting

Figure 6: The visualization of training data distributions for online continual learning under different settings. The horizontal axis represents the time step or batch, and the vertical axis denotes the class index. Classes sorted by their appearance time within each distribution.

Figure 6 illustrates the distribution of training data across different online continual learning settings, with classes sorted by their appearance order within each group. As shown in Figure 6a, the online CIL setting is consistent with the CIL framework, where there is no overlap between classes across different tasks. The Gaussian schedule, depicted in Figure 6b, builds upon this by assuming that samples should adhere to a Gaussian distribution. The Si-Blurry setting divides all classes into two groups, where $N\%$ of the classes are selected as disjoint classes and the remaining 100-$N\%$ of the classes are selected as blurry$M$ classes, with $M$ representing the blurry level Koh et al. (2022). The number of disjoint and blurry classes are randomly assigned to each task, as shown in Figure 6c. The blurry classes of each task may overlap, thereby obscuring the clear boundaries between tasks. Finally, our proposed MiD-Blurry setting, illustrated in Figure 6d, consists of three types of distributions: disjoint classes, Gaussian classes, and decay classes.

## A.2 ONLINE CONTINUAL LEARNING ON MID-BLURRY SETTING

Figure 7 illustrates the performance of various methods during online continual learning across datasets. Notably, our proposed SIT strategy consistently outperformed other methods on CIFAR-100 and TinyImageNet. In the ImageNet-R dataset, SIT-CLIP exhibited performance comparable to MoEAdapter, despite the latter having a parameter count ten times greater than that of SIT-CLIP. Additionally, it is important to highlight that DualPrompt and MVP demonstrated significantly lower performance on ImageNet-R. This performance drop can be attributed to the fact that their backbone networks are pre-trained on natural images, making them less adaptable to the distribution of data in ImageNet-R.

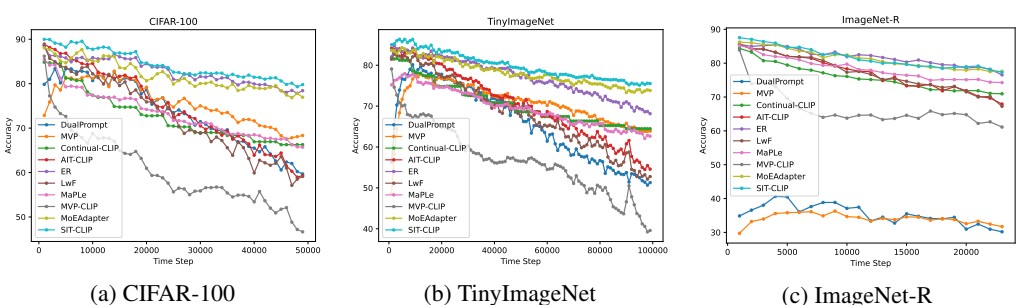

| (a) CIFAR-100 | (b) TinyImageNet | (c) ImageNet-R |

Figure 7: Performance comparison of online continual learning on MiD-Blurry setting.

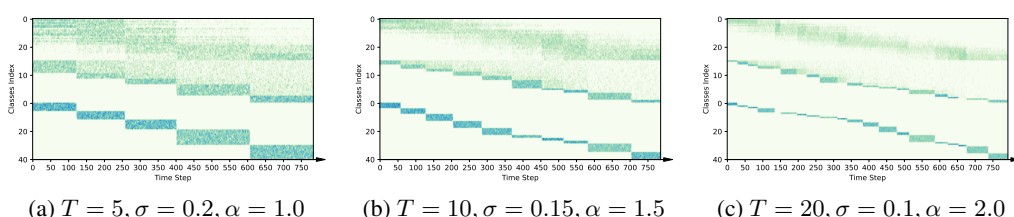

| (a) $T = 5, \sigma = 0.2, \alpha = 1.0$ | (b) $T = 10, \sigma = 0.15, \alpha = 1.5$ | (c) $T = 20, \sigma = 0.1, \alpha = 2.0$ |

Figure 8: The visualization of different training data distributions.

### A.3 ONLINE CONTINUAL LEARNING PERFORMANCE UNDER DIFFERENT DATA DISTRIBUTIONS

This experiment aims to compare online continual learning performance under different data distributions within the Mid-Blurry setting, as illustrated in Figure 8. Our proposed SIT-CLIP method achieved the best performance in the relatively stable scenarios of $T = 5$ and $T = 10$. However, in the more unstable setting of $T = 20$, MoEAdapter, with its larger parameter count, slightly outperformed SIT-CLIP. Notably, in the stable scenarios of $T = 5$ and $T = 10$, the performance of Continual-CLIP is even lower than that of classification-based methods, indicating potential limitations in its adaptability compared to our approach.

Table 4: Comparison of online continual learning performance under different data distributions, where $T$ represents the number of tasks in the MiD-Blurry setting, $\sigma$ is the standard deviation of the Gaussian classes, and $\alpha$ is the coefficient for the decay classes. In all experiments, Gaussian classes and decay classes each account for 30% of the total classes.

| Method | $T = 5, \sigma = 0.2, \alpha = 1.0$ | | $T = 10, \sigma = 0.15, \alpha = 1.5$ | | $T = 20, \sigma = 0.1, \alpha = 2.0$ | |
|---|---|---|---|---|---|---|
| | $\mathcal{A}_{auc}$ | $\mathcal{A}_{last}$ | $\mathcal{A}_{auc}$ | $\mathcal{A}_{last}$ | $\mathcal{A}_{auc}$ | $\mathcal{A}_{last}$ |
| DualPrompt | 72.88±0.55 | 61.74±1.71 | 73.63±1.10 | 58.31±2.34 | 71.29±0.34 | 55.48±2.91 |
| MVP | 74.83±0.16 | 70.83±0.62 | 75.70±0.93 | 67.70±1.46 | 71.25±0.54 | 61.95±2.12 |
| Continual-CLIP | 71.13±0.57 | 66.27±0.00 | 72.66±0.94 | 66.27±0.00 | 74.69±0.21 | 66.27±0.00 |
| AIT-CLIP | 74.96±0.09 | 65.28±0.05 | 74.52±0.45 | 59.52±1.17 | 69.23±0.57 | 47.27±1.56 |
| MaPLe | 72.74±0.01 | 68.75±1.46 | 73.25±0.68 | 66.18±1.64 | 74.91±0.69 | 65.55±0.72 |
| MoEAdapter | 81.48±0.72 | 79.54±0.69 | 81.88±0.17 | 77.39±0.27 | **81.58±0.34** | **75.78±0.16** |
| **SIT-CLIP (Ours)** | **81.84±0.31** | **79.94±0.89** | **84.34±0.56** | **79.47±0.52** | 81.47±0.05 | 74.51±0.30 |

Table 5: Performance comparison of online continual learning methods on Si-Blurry setting with disjoint class ratio $N = 50$, blurry level $M = 10$ and task number $T = 5$. Note that the buffer size of the second group of methods is 2000, and the other methods are rehearsal-free.

| Method | CIFAR-100 | | TinyImageNet | | ImageNet-R | |
|---|---|---|---|---|---|---|
| | $\mathcal{A}_{auc}$ | $\mathcal{A}_{last}$ | $\mathcal{A}_{auc}$ | $\mathcal{A}_{last}$ | $\mathcal{A}_{auc}$ | $\mathcal{A}_{last}$ |
| Finetuning | 19.71±3.39 | 10.42±4.92 | 15.50±0.74 | 10.42±4.92 | 7.51±3.94 | 2.29±0.85 |
| Linear Probing | 49.69±6.09 | 23.07±7.33 | 42.15±2.79 | 21.97±6.43 | 29.24±1.26 | 16.87±3.14 |
| ER | 69.86±4.08 | 71.81±0.69 | 66.75±1.13 | 55.07±1.28 | 45.74±1.35 | 38.13±0.32 |
| EWC++ | 47.75±5.35 | 46.93±1.44 | 64.92±1.21 | 53.04±1.53 | 30.20±1.31 | 21.28±1.88 |
| RM | 53.27±3.00 | 65.51±0.55 | 47.26±1.13 | 44.55±0.37 | 27.88±1.29 | 24.25±0.99 |
| CLIB | 71.53±2.61 | 72.09±0.49 | 65.47±0.76 | 56.87±0.54 | 42.69±1.30 | 35.43±0.38 |
| MVP-R | 78.65±3.59 | **84.42±0.44** | **80.67±0.75** | 74.34±0.32 | 52.47±1.45 | 50.54±2.08 |
| LwF | 55.51±3.49 | 36.53±10.96 | 49.00±1.52 | 27.47±7.59 | 31.61±1.53 | 20.62±3.67 |
| L2P | 57.08±4.43 | 41.63±12.73 | 52.09±1.92 | 35.05±5.73 | 29.65±1.63 | 19.55±4.78 |
| DualPrompt | 67.07±4.16 | 56.82±3.49 | 66.09±2.00 | 48.72±3.41 | 40.11±1.27 | 29.24±4.63 |
| MVP | 68.10±4.91 | 62.59±2.38 | 68.95±1.33 | 52.78±2.08 | 40.60±1.21 | 31.96±3.07 |
| Continual-CLIP | 69.57±1.05 | 66.26±0.02 | 71.55±0.97 | 65.18±0.03 | 76.63±0.52 | 71.12±0.34 |
| **SIT-CLIP (Ours)** | **81.77±1.80** | 80.80±1.26 | 80.64±1.18 | **75.08±1.09** | **84.15±0.38** | **79.35±0.35** |

## A.4 ONLINE CONTINUAL LEARNING ON SI-BLURRY SETTING

In this experiments, we utilized the Si-Blurry setting to evaluate our proposed method on CIFAR-100, TinyImageNet, and ImageNet-R datasets. Specifically, we set the disjoint class ratio $N = 50$, blurry level $M = 10$ and task number $T = 5$. Our method is compared with several state-of-the-art online continual learning approaches, including replay-based methods ER Rolnick et al. (2019), RM Bang et al. (2021), and CLIB Koh et al. (2022), regularization-based method LwF Li & Hoiem (2017), a combination of replay-regularization in EWC++ Kirkpatrick et al. (2017), and prompt-based methods L2P Zhou et al. (2022b), DualPrompt Wang et al. (2022), MVP and MVP-R Moon et al. (2023). Additionally, we considered Continual-CLIP, which leverages the zero-shot learning capability of CLIP model, and set finetuning and linear probing as our lower-bound benchmarks. All methods employ a pre-trained Vision Transformer (ViT-B/16) as the backbone model. For replay-based methods ER, RM, CLIB, and MVP-R, the buffer size is consistently set to 2000 to ensure a fair comparison. The results, as depicted in Table 5, reveal that CLIB and MVP, designed specifically for boundary-blurred scenarios, perform well on general datasets like CIFAR-100 and TinyImageNet. However, even with replay data, these classical classification models only marginally outperform Continual-CLIP and significantly underperform on the domain-shift designed ImageNet-R. Furthermore, we found that applying the SIT strategy to tune CLIP model via PET can substantially enhance the performance of CLIP model without the need for replay, knowledge distillation, or other auxiliary techniques. Our findings underscore the effectiveness of our proposed method, which not only matches but also surpasses the performance of existing methods, highlighting its potential as a robust solution for continual learning in blurred and dynamic environments.

## A.5 CLASS-INCREMENTAL LEARNING

Table 6 presents the experimental results in the class-incremental learning setting. We compare our approach with several state-of-the-art class-incremental learning methods, where the classification-based methods include iCaRL Rebuffi et al. (2017), BiC Wu et al. (2019), WA Zhao et al. (2020) and DER Yan et al. (2021), all of which use ResNet18 as backbone network. In contrast, PRAKA Shi & Ye (2023), AFC Kang et al. (2022), DyTox Douillard et al. (2022), and HFC Dong et al. (2023) utilize ViT-B/16 as backbone. Continual-CLIP, ZSCL Zheng et al. (2023), MoEAdapter Yu et al. (2024), and SIT-CLIP are VLM-based methods that also employ ViT-B/16 as backbone. The results for the compared methods here are derived from the reports in their respective papers. For our proposed SIT-CLIP, we maintain an online training approach, i.e. training for only one epoch on each task. It is evident that our SIT-LoRA method outperforms all other methods in both task

settings. Specifically, in the more challenging 20-task setting, SIT-LoRA achieves an impressive $\mathcal{A}_{avg}$ of 86.45% and an $\mathcal{A}_{last}$ of 78.67%. These results demonstrate the efficacy of our SIT strategy, showcasing its potential to compete with or even surpass established CIL methods while adhering to more stringent online learning constraints.

Table 6: Performance comparison of class-incremental learning methods on CIFAR-100 benchmark.

| Method | 10 tasks | | 20 tasks | |
|---|---|---|---|---|
| | $\mathcal{A}_{avg}$ | $\mathcal{A}_{last}$ | $\mathcal{A}_{avg}$ | $\mathcal{A}_{last}$ |
| iCaRL | 65.27 | 50.74 | 61.20 | 43.74 |
| BiC | 68.80 | 53.54 | 66.48 | 47.02 |
| WA | 69.46 | 53.78 | 67.33 | 47.31 |
| DER | 74.64 | 64.35 | 73.98 | 62.55 |
| PRAKA | 68.86 | - | 65.86 | - |
| DyTox | 74.10 | 62.34 | 71.62 | 57.43 |
| AFC | 75.50 | - | 70.30 | - |
| HFC | 86.30 | - | 85.50 | - |
| Continual-CLIP | 75.17 | 66.72 | 75.95 | 66.72 |
| ZSCL | 82.15 | 73.65 | 80.39 | 69.58 |
| MoEAdapter | 85.21 | 77.52 | 83.72 | 76.20 |
| **SIT-CLIP (Ours)** | **86.88±0.34** | **80.38±0.57** | **86.45±0.40** | **78.67±0.70** |

## A.6 MORE DETAILED MEMORY STABILITY EVALUATION.

In this section, we assess the zero-shot performance of various VLM-based online continual learning methods across different datasets. Table 7 presents the results of zero-shot evaluations conducted after online continual learning on CIFAR-100 within the MiD-Blurry setting. The findings indicate that our SIT-CLIP method achieves the highest average accuracy of 84.34%, outperforming several existing approaches. Notably, the ER method also shows strong performance, primarily due to its effective memory stability and replay data strategy. In contrast, the Continual-CLIP baseline exhibits lower performance across multiple targets, highlighting the advantages of our approach.

Table 8 summarizes the zero-shot performance results on the more challenging ImageNet-R dataset. Here, SIT-CLIP again achieves a commendable accuracy of 81.65%, demonstrating competitive performance with the ER and MoEAdapter methods. The findings suggest that SIT-CLIP retains a robust capability to generalize across diverse datasets, effectively learning from cross-domain data. The results highlight the memory stability of the methods evaluated, with SIT-CLIP showing minimal degradation compared to the non-finetuned baseline, reinforcing the effectiveness of leveraging fine-grained and cross-domain datasets in enhancing model performance.

Table 7: Comparison of zero-shot performance after online continual learning in CIFAR-100 with the MiD-Blurry setting. The best results are in bold, and the second-best results are underlined. This experiment aims to evaluate the memory stability of the model after online continual learning on general datasets.

| Method | CIFAR-100 | | Targets | | | | | | | | | |
|---|---|---|---|---|---|---|---|---|---|---|---|---|
| | $\mathcal{A}_{auc}$ | $\mathcal{A}_{last}$ | Flowers102 | OxfordPet | EuroSAT | Food101 | SUN397 | FGVCAircraft | CUB200 | StanfordCars | Caltech101 | Average |
| *Continual-CLIP* | *72.66±0.94* | *66.27±0.00* | *65.86* | *85.33* | *40.24* | *86.34* | *61.54* | *21.63* | *52.60* | *57.08* | *87.95* | *62.06* |
| AIT-CLIP | 74.52±0.45 | 59.52±1.17 | 57.42±3.05 | 80.50±0.95 | 35.25±2.91 | 81.91±0.98 | 61.04±1.42 | 18.86±1.68 | 40.43±1.11 | 54.04±2.76 | 91.02±1.36 | 57.83±0.20 |
| ER | 82.77±0.37 | 78.00±1.23 | 60.91±1.77 | 80.39±1.48 | 34.23±2.51 | 80.99±1.03 | 64.25±0.14 | 20.53±0.74 | 43.31±1.32 | **55.08±1.53** | 92.76±0.29 | 59.16±0.47 |
| LwF | 72.60±0.65 | 57.60±2.87 | 52.38±3.54 | 77.90±2.44 | 30.36±3.37 | 79.41±1.89 | 61.32±1.17 | 18.12±0.65 | 38.60±2.26 | 54.37±0.84 | 90.06±1.95 | 55.84±0.88 |
| MaPLe | 73.25±0.68 | 66.18±1.64 | 60.06±0.09 | 77.94±3.35 | **38.16±1.90** | **84.38±1.54** | 61.02±0.88 | 16.57±0.61 | 43.77±1.66 | 50.59±3.20 | 91.41±0.27 | 58.21±0.88 |
| MVP-CLIP | 60.44±0.93 | 45.87±0.33 | 62.60±0.68 | 81.93±1.09 | 30.97±0.93 | 82.40±0.29 | 59.77±0.50 | **21.26±0.11** | 45.28±0.90 | 53.75±0.19 | 91.77±0.26 | 58.86±0.13 |
| MoEAdapter | 81.88±0.17 | 77.39±0.27 | 63.60±2.84 | 81.11±0.67 | 32.87±3.92 | 79.78±1.20 | 63.81±0.55 | 18.50±1.15 | 42.55±1.86 | 54.89±1.16 | 93.10±0.18 | 58.91±1.29 |
| SIT-CLIP | **84.34±0.56** | **79.47±0.52** | **63.12±1.68** | **82.88±1.55** | 33.32±2.62 | 82.68±0.60 | **65.27±0.70** | 19.02±0.96 | **45.30±1.33** | 55.00±1.64 | **93.91±0.43** | **60.06±0.51** |

Table 8: Comparison of zero-shot performance after online continual learning in ImageNet-R with the MiD-Blurry setting. The best results are in bold, and the second-best results are underlined. This experiment aims to evaluate the memory stability of the model after online continual learning on cross-domain datasets.

| Method | ImageNet-R | | Targets | | | | | | | | | |
|---|---|---|---|---|---|---|---|---|---|---|---|---|
| | $\mathcal{A}_{auc}$ | $\mathcal{A}_{last}$ | Flowers102 | OxfordPet | EuroSAT | Food101 | SUN397 | FGVCAircraft | CUB200 | StanfordCars | Caltech101 | Average |
| *Continual-CLIP* | *75.76±0.29* | *71.06±0.35* | *65.86* | *85.33* | *40.24* | *86.34* | *61.54* | *21.63* | *52.60* | *57.08* | *87.95* | *62.49* |
| AIT-CLIP | 76.99±0.68 | 65.21±1.32 | 58.75±0.71 | 76.22±0.84 | 31.69±2.06 | 80.59±0.03 | 61.02±1.56 | 19.22±0.27 | 39.13±1.98 | 52.39±0.85 | 87.03±0.55 | 56.44±0.56 |
| ER | 80.76±0.19 | 76.05±0.15 | 61.75±0.83 | 80.71±1.49 | 28.61±3.05 | 82.22±0.93 | 64.17±0.10 | 19.44±1.14 | 43.28±1.20 | 53.38±0.53 | 90.42±1.27 | 58.71±0.31 |
| LwF | 77.04±0.58 | 66.24±1.24 | 61.07±2.40 | 76.22±1.94 | 31.90±0.85 | 81.04±0.55 | 61.75±2.32 | 18.68±1.17 | 40.36±1.07 | 53.07±1.46 | 88.06±1.19 | 57.18±0.95 |
| MaPLe | 78.33±0.38 | 74.15±0.41 | 61.91±0.31 | 84.65±0.33 | 34.22±2.93 | 85.31±0.89 | 63.92±0.20 | 18.89±0.29 | 48.06±0.99 | 54.79±0.98 | 91.40±1.12 | 60.92±0.37 |
| MVP-CLIP | 66.53±1.14 | 57.79±1.18 | 63.47±0.41 | 81.05±0.61 | 30.41±0.66 | 79.56±0.01 | 58.52±0.77 | 19.31±0.02 | 44.54±0.10 | 50.43±0.02 | 91.33±0.33 | 58.09±0.12 |
| MoEAdapter | 81.30±0.12 | 76.88±0.16 | 62.43±0.76 | 82.07±1.28 | 25.77±2.13 | 81.25±0.71 | 63.69±0.66 | 19.97±1.30 | 43.84±0.88 | 55.00±1.40 | 91.01±0.99 | 58.78±0.54 |
| SIT-CLIP | 81.65±0.18 | 77.53±0.65 | 62.15±0.90 | 81.75±1.41 | 30.40±5.36 | 82.56±1.29 | 63.90±1.31 | 21.12±0.86 | 44.09±0.56 | 54.85±2.38 | 90.19±0.23 | 59.40±0.99 |

Table 9: Comparison of Different Loss Functions. This experiment is performed on CUB200, StanfordCars, FGVCAircraft with the MiD-Blurry setting.

| Method | $\mathcal{A}_{auc}$ | $\mathcal{A}_{last}$ | Flowers102 | OxfordPet | EuroSAT | Food101 | SUN397 | Caltech101 | Average |
|---|---|---|---|---|---|---|---|---|---|
| *Continual-CLIP* | *57.46* | *48.72* | *65.86* | *85.33* | *40.24* | *86.34* | *61.54* | *87.95* | *70.50* |
| AIT-CLIP | 59.32 | 43.69 | 64.25 | 84.31 | 26.88 | 83.39 | 59.79 | 84.83 | 66.67 |
| Focal Loss | 62.54 | 46.16 | 66.35 | 86.68 | 28.46 | 84.34 | 61.10 | 88.81 | 69.29 |
| LDAM | 58.79 | 43.92 | 68.47 | 85.60 | 29.37 | 83.90 | 61.39 | 87.47 | 69.37 |
| $\lambda_A = 0.5$ | 58.44 | 31.82 | 56.94 | 82.07 | 23.38 | 74.55 | 53.62 | 72.00 | 60.43 |
| $\lambda_A = C_{batch}/C_{seen}$ | 64.95 | 55.84 | 64.38 | 85.05 | 30.44 | 79.38 | 58.48 | 79.59 | 66.22 |
| SIT-CLIP | 65.86 | 57.05 | 67.59 | 86.68 | 38.32 | 79.49 | 63.66 | 91.15 | 70.24 |

## A.7 ABLATION STUDY

### A.7.1 COMPARISON OF DIFFERENT LOSS FUNCTIONS

In this ablation study, we evaluate the effectiveness of removing asymmetric negative samples by comparing different loss functions. Specifically, Focal Loss and LDAM are classic loss functions designed for imbalanced learning. We also considered introducing weights into the loss function as $L = L_S + \lambda_A L_A$, with $C_{batch}$ representing the number of classes in the current batch and $C_{seen}$ representing the total number of seen classes. As shown in the table, although advanced logit adjustment strategies yield incremental improvements under continual learning settings, they introduce additional complexity. In contrast, SIT demonstrates a favorable balance between simplicity and effectiveness. Thus, we believe our approach provides an optimal trade-off between performance and complexity.

### A.7.2 ANALYSIS OF THE PEFT METHODS ON CLIP MODEL

In this experiment, we conducted a comparative analysis of various PEFT methods to evaluate their impact on performance and generalizability in online continual learning. Specifically, we performed online continual learning on the CIFAR-100 dataset, while conducting zero-shot evaluations on CUB200, StanfordCars, FGVCAircraft, Flowers102, OxfordPet, EuroSAT, Food101, SUN397, and Caltech101. Due to space constraints, only the average results of the zero-shot evaluations are presented in Tables 10. As evident from the results, LoRA achieved the best performance among the evaluated methods. MVP-CLIP, which is based on prompt tuning, exhibited minimal improvements in online continual learning. Both Adapter and MoEAdapter demonstrated strong performance in this context; notably, Adapter requires fewer parameters, whereas the MoE structure in MoEAdapter aids in maintaining the stability of knowledge acquired during pre-training.

Table 12: Comparison the impact of batch size in online continual learning.

| Method | $\mathcal{A}_{\text{auc}}$ | $\mathcal{A}_{\text{last}}$ |
|---|---|---|
| *0* | *71.55±0.97* | *65.18±0.03* |
| 8 | 74.44±0.90 | 63.85±0.87 |
| 16 | 78.29±0.29 | 70.06±1.33 |
| 32 | 80.20±0.50 | 73.58±0.31 |
| 64 | 80.91±0.46 | 74.91±0.30 |
| 128 | 80.83±0.52 | 75.50±0.22 |

Table 10: Comparative analysis of PEFT methods.The best results are in bold. This experiment aims to evaluate the effects on performance and generalizability in online continual learning.

| Method | #P | #TP | $\mathcal{A}_{\text{auc}}$ | $\mathcal{A}_{\text{last}}$ | Zero-shot |
|---|---|---|---|---|---|
| Adapter | 1.98 M | 151.60 M | 81.06±0.19 | 76.66±1.04 | 56.92±0.16 |
| LoRA | **0.37 M** | 149.99 M | **84.34±0.56** | **79.47±0.52** | **60.06±0.51** |
| MoEAdapter | 4.03 M | 153.65 M | 81.80±0.13 | 77.81±0.53 | 58.68±0.54 |
| MVP-CLIP | 0.48 M | 150.10 M | 71.99±0.59 | 63.87±1.32 | 59.17±0.09 |

Table 11: Comparison of training and inference efficiency of different PEFT methods. This experiment conducted on a single NVIDIA 3090 GPU.

| Method | # TP | # T | FLOPs | Train | Test |
|---|---|---|---|---|---|
| Continual-CLIP | 0.00 M | 149.62 M | 46.15 G | N/A | 365.92±1.38 item/s |
| Full FT | 149.62 M | 149.62 M | 46.15 G | 30.17±0.00 item/s | 366.27±1.22 item/s |
| LORA | 149.99 M | 0.37 M | 47.44 G | 35.24±0.17 item/s | 345.82±1.34 item/s |
| Adapter | 151.60 M | 1.98 M | 46.74 G | 32.62±0.07 item/s | 335.46±1.56 item/s |
| MoEAdapter | 153.65 M | 4.03 M | 46.74 G | 31.12±0.44 item/s | 310.45±1.86 item/s |

### A.7.3 ANALYSIS OF THE IMPACT OF BATCH SIZE IN ONLINE CONTINUAL LEARNING.

The objective of this experiment is to determine how variations in batch size affects the efficiency and effectiveness of the learning process in an online continual learning scenario. We conducted a series of experiments with batch sizes ranging from 8 to 128 and use the Si-Blurry setting on the tinyImagenet dataset. In addition, batchsize is set to 0 to indicate no online learning, i.e., Continual-CLIPThengane et al. (2022). The results, as indicated in Table 12, demonstrate that the overall impact of batch size on online continual learning performance is relatively minor, with the $\mathcal{A}_{\text{auc}}$ differing by only 6.39% between the smallest and largest batch sizes tested. Notably, the final performance in online continual learning converges when the batch size exceeds 16, suggesting that beyond this point, increasing the batch size does not yield significant improvement in performance.

