# OpenReview forum: "CLIP model is an Efficient Online Continual Learner"
_ICLR.cc/2025/Conference — Submitted to ICLR 2025_

### Official Review · Reviewer_Dpce · 2024-11-01

**Soundness:** 2
**Presentation:** 3
**Contribution:** 2
**Rating:** 3
**Confidence:** 4

**Summary:**

The paper shows the performance of CLIP in Online Continual Learning under parameter-efficient adaptations. A regularization method for mitigating the bias toward asymmetric examples is also discussed. The proposed method was tested on popular Continual Learning with the aid of a novel setup that simulates the distribution of non-stationary data streams in a more challenging way compared to previous methods, called MiD-Blurry. The proposed model is tested against the proposed benchmark and compared with both CLIP-based methods and standard classification models used in class-incremental learning.

**Strengths:**

The authors make an effort to identify some flaws in the base CLIP behavior when facing online continual learning data streams, by performing a gradient-based analysis to detect a systematic bias toward newer classes in the training phase. Furthermore, the authors understood the limitations of current benchmarks for evaluating performance under Online CL circumstances and introduced a setup to handle data streams by providing an additional challenge focused on non-stationarity. Understanding the limitations of current benchmarks is a crucial point, especially in Online CL, where real-case applications require facing complex dynamic environments.

**Weaknesses:**

Although the proposed method successfully challenges some intrinsic flaws of contrastive learning pre-training methods by introducing the SIT strategy, I don't feel like it is a relevant innovation. Regularizing the loss by adjusting for asymmetric examples has been already accounted for (see Manco et al (2022) - Contrastive Audio-Language Learning for Music, and Zolfaghari et al (2021) - CrossCLR Cross-modal Contrastive Learning for Multi-modal Video Representations).

A similar issue is raised for the parameter-efficient adaptation of CLIP. I think nowadays, prompt-, prefix-, and adapters-based fine-tuning are no longer a relevant design choice. Several models can benefit from these adaptations in the context of vision-language modeling (e.g. Flava, BEiT3), and the authors don't provide any insight into why CLIP was selected in place of other multimodal models leveraging contrastive pre-training.

Despite the effort to mitigate the lack of complexity of current CL benchmarks, the proposed MiD-Blurry still deals with standard continual learning datasets, which present non-complex data with high semantic correlation In general, don't see any additional challenge other than the one added by the proper online CL criteria. These datasets can't resemble real-case scenarios where the online CL setting should be first employed. From a benchmarking perspective, methods like DualPrompt that show amazing performance on the standard CL benchmark can under-perform under the circumstances of online CL likely due to the lack of a proper hyper-parameter tuning or learning pace of the prompts. I think a more suitable benchmark for online CL is needed, and the proposed setup might be not that relevant for facing the issue of the lack of a proper benchmark.

**Questions:**

In line 138 the authors claim that the main limitation of mentioned CL methods is that they are closed-ended since they handle a pre-defined classifier dimension. This is not true, since in practice it's easy to use an incremental classifier that adds new heads following the same CL adaptation of the parameters, without knowing a priori the total number of classes in the whole experiences set. The incremental classifier adaptation can be safely used in online continual learning in principle, although it can be quite underperforming of course. This point is discussed again in line 239. Again, I don't see a theoretical limitation in adjusting the architecture when novel classes arrive when the architecture to be adjusted is just an additional incremental classification head, and not the main backbone (the feature extractor). When the set of total classes encountered starts increasing in size significantly, CLIP also can have efficiency-related challenges, since, for a given image batch on a given experience, you have to process the whole set of class-label prompted sentences on the text encoder, which posits a computational burden for huge number of experiences. Therefore, if the argument is solely based on efficiency, a computational efficiency analysis should be provided.

---

> ### Author Response · Authors · 2024-11-20
>
> **Q1. Definition of asymmetric negative example and comparison with similar methods**
> We appreciate your suggestion to compare our method with [1] and [2]. In our work, asymmetric negative examples are defined as text labels without corresponding image samples within a batch. As discussed in Section 3.3, these samples, while useful for retaining knowledge of old classes, adversely impact the learning of new classes.
>
> While [1] and [2] address cross-modal contrastive learning, their methods assume symmetry between positive and negative examples, unlike the task-agnostic online continual learning setup we address. Additionally, our experiments indicate that logit adjustment techniques used in [1] and [2] do not yield significant improvements in our setting. This prompted us to adopt a simpler yet effective approach of excluding asymmetric negative samples. We will revise the manuscript to clarify these distinctions and highlight the unique challenges addressed by our method.
>
> **Q2. Choice of CLIP as the Vision-Language Model**
> Thank you for your feedback regarding the choice of CLIP over models like Flava and BEiT3. While these models offer strong multimodal capabilities, they are larger and more complex than CLIP, making them less practical for the resource-constrained online continual learning scenarios we target. CLIP’s simplicity and proven effectiveness across various tasks make it a better baseline for fair benchmarking.
>
> Moreover, our ablation studies confirm that the proposed method is not dependent on any specific parameter-efficient fine-tuning (PEFT) technique, demonstrating generalizability. This underscores that our method is not merely a "trick" tailored for CLIP. We will explicitly include this rationale in the revised manuscript to better justify our choice.
>
> **Q3. Complexity of MiD-Blurry Online CL Setting**
> We recognize the need for realistic benchmarks for online continual learning. The MiD-Blurry setup addresses this by introducing dynamic and imbalanced class distributions, which better resemble real-world online CL scenarios than classical benchmarks. As described in Section 3.1, MiD-Blurry challenges models with continually shifting class distributions, adding complexity beyond that of Si-Blurry or standard CIL setups.
>
> To further illustrate its relevance, our experiments in Table 2 demonstrate the model's adaptability across distinct fine-grained datasets like CUB200, StanfordCars, and FGVCAircraft, which have minimal semantic overlap. This setup tests the model’s ability to adapt to diverse, fine-grained tasks. We will enhance our discussion to emphasize these challenges.
>
> **Q4. Suitability of Current Online CL Benchmarks**
> We appreciate your point about the limitations of existing benchmarks for online CL. To fairly evaluate our method's performance, we compared it to [3]'s task-agnostic online CL setup, including both the Si-Blurry and our proposed MiD-Blurry settings. In [3], the DualPrompt method showed competitive results, and we include it as a comparison in our work as well. However, we explicitly note that DualPrompt was not designed for task-agnostic online CL and uses a non-VLM architecture. We have clearly marked its results in Table 1 for transparency.
>
> **Q5. Clarification on Line 138 and Incremental Classifier Limitations**
> We appreciate your suggestion to compare with more non-VLM continual learning methods. In addition to DualPrompt and MVP, we are working to reproduce [4] and include that comparison as well.
>
> We understand that adding incremental classification heads is a valid approach, as you rightly pointed out, it often underperforms and requires careful initialization. In contrast, VLM-based methods, by matching image-text pairs, effectively mitigate these issues and can leverage semantic information in language. Furthermore, open-world classification is a good property of VLMs, which is why CLIP is our preferred choice—not the only choice.
>
> While we did not emphasize efficiency in the manuscript, we would like to clarify that text encoding can be precomputed and used as a fixed classifier, which is computationally equivalent to non-VLM methods.
>
> ---
>
>  [1] Contrastive Audio-Language Learning for Music. ISMIR 2022
>
>  [2] CrossCLR Cross-modal Contrastive Learning for Multi-modal Video Representations. ICCV 2021
>
>  [3] Online Continual Learning on Class Incremental Blurry Task Configuration with Anytime Inference. ICLR 2022
>
>  [4] Anytime Continual Learning for Open Vocabulary Classification. ECCV 2024

---

> > ### Author Response · Authors · 2024-12-02
> >
> > Dear Reviewer Dpce,
> >
> > Thank you again for your valuable feedback. We hope that the explanations provided address your concerns regarding the SIT method, its limitations, and the associated experiments. We would greatly appreciate it if you could let us know whether our responses have clarified your doubts or if there are any other aspects you would like to discuss further.
> >
> > We are eager to ensure that all points are fully addressed and would be happy to provide additional clarifications or discuss any remaining questions.
> >
> >
> > Best regards,
> >
> > The Authors

---

### Official Review · Reviewer_p39U · 2024-11-02

**Soundness:** 3
**Presentation:** 4
**Contribution:** 2
**Rating:** 5
**Confidence:** 5

**Summary:**

This paper focuses on online continual learning, which addresses the challenge of learning from continuous, non-stationary data streams. Leveraging CLIP's open-world classification ability, online continual learning frameworks enable a flexible learning process without the constraint of class labels. To further improve the ability of CLIP in online continual learning without being affected by the catastrophic forgetting problem brought by finetuning, this paper proposes the symmetric image-text (SIT) strategy by removing asymmetric negative texts in online learning. In addition, this paper also introduces a challenging online continual learning setting named MiD-Blurry, which mixes multiple training data distributions to simulate real-world scenarios better. Results on multiple image classification datasets show that the proposed method performs preferably against previous state-of-the-art methods with minimal trainable parameters under both the Si-Blurry setting and the proposed MiD-Blurry setting.

**Strengths:**

1. The paper analyzes the problem of catastrophic forgetting during parameter efficient fine-tuning (PEFT) by dividing negative text samples into two parts: symmetric negative sample (SNS), negative text features corresponding to the classes of the images in the current batch, and asymmetric negative sample (ANS), negative text features corresponding to the classes of the images that not in the current batch. The gradient norm visualization directly shows the cause of catastrophic forgetting and the reason for the efficiency of the proposed method, SIT. This analysis is straightforward and convincing.

2. The proposed MiD-Blurry setting simulates the real-world scenario in which the classes suddenly appear and gradually fade away. The setting aligns with the basic idea of online open-world continual learning.

3. The selected datasets for evaluation are comprehensive and diverse, especially for evaluating zero-shot performance after online continual learning.

**Weaknesses:**

1. Figures:

Figure 1 right cannot really show the idea of SIT. I found the direction of the arrows confusing.

In Figure 4, the representation is hard to see clearly and follow.

2. Method:

The overall method is straightforward. However, I think removing the contrastive learning between current images and the negative samples of classes on other batches/datasets is against the idea of online continual learning in the open-world setting. SIT can give you good performance when you evaluate on each dataset. However, this training manner cannot help you distinguish the positive images and negative texts across different datasets. This is important in open-world classification, a good property of using CLIP.

This shortage can also be seen in the experiment results. After online learning on some general datasets, the zero-shot performance on Food 101 drops a lot. This is because compared to other fine-grained datasets, Food 101 shares more common images/classes with general datasets like ImageNet and Caltech 101, so the SIT training strategy will affect the performance of Food 101. However, it's good that the authors include the Food 101 dataset in the comparison rather than drop it because of the inferior performance.

**Questions:**

For the evaluation of zero-shot performance on fine-grained datasets, what if you add all previously learned classes on general datasets into the text labels for classification? I assume due to the lack of training between image samples and text samples from other batches, the fine-tuned CLIP model will lose some ability for open-world classification, i.e., the ability to distinguish images from labels of other datasets. A similar evaluation setting can be found in these two papers:

Continual Learning in Open-vocabulary Classification with Complementary Memory Systems. TMLR 2024

Anytime Continual Learning for Open Vocabulary Classification. ECCV 2024

Which could also be baselines to evaluate against.

---

> ### Author Response · Authors · 2024-11-20
>
> **Q1. Clarification of Figure 1 and SIT Visualization**
> We appreciate the feedback on Figure 1 and acknowledge the potential confusion regarding the visualization of SIT. The figure's purpose is to illustrate the conceptual distinction between traditional classification and CLIP-based approaches. Specifically:
> - Upper Part: The upper panel demonstrates how a classification model computes logits for prediction.
> - Lower Part: The lower panel shows CLIP extracting features for both images and classes separately to perform matching. For SIT, the dashed arrow indicates that asymmetric class labels are not used during training at step t, highlighting that negative samples from prior classes are excluded.
> We will revise Figure 1 to make these distinctions clearer, including relabeling arrows and adding explanatory annotations.
>
> **Q2. Representation in Figures 4 and 5:**
> We acknowledge that Figures 4 and 5 can be challenging to interpret. In these figures, we reduced the features to the same lower-dimensional space using PCA to facilitate the comparison of differences between methods. In Figure 4, we can observe that Continual-CLIP is effective at distinguishing data from different domains (CIFAR-100 and Flowers102), but it struggles to differentiate data within the same domain. SIT, on the other hand, significantly improves the matching of seen data (CIFAR-100) while maintaining the ability to distinguish unseen data (Flowers102). However, AIT excessively pushes image features away from text features, which undermines the model's ability to differentiate unseen data.
>
> We will work on improving the clarity of these visualizations by adjusting the color schemes, labeling, and providing additional context to guide readers through the figures.

---

> > ### Author Response · Authors · 2024-11-20
> >
> > **Q3. Reason for removing asymmetric contrastive samples**
> > The removal of asymmetric negative samples (ANS) is based on their disproportionate impact on learning dynamics. Section 3.3 explains how the loss decomposition highlights the role of $L_{A}$, which penalizes proximity between current image features and past class representations. While $L_{A}$ aids memory retention, its excessive influence hinders learning for new classes as the number of seen classes increases. As shown in Figure 3, gradients from ANS dominate parameter updates, leading to a bias toward older classes. This is reflected in Figure 4, where SIT balances domain alignment while maintaining generalization better than AIT. Considering that PEFT freezes the pretrained model and introduces a small and trainable adapter, which essentially functioning as a parameter isolation technique, we believe that simply removing ANS is sufficient.
> >
> > To validate this point, we conducted experiments under the same settings as in Table 2, comparing SIT with other loss functions. Among these, Focal Loss [3] and LDAM [4] are classic loss functions designed for imbalanced learning. We also considered introducing weights into the loss function as $L=L_{S}+\lambda_{A}L_{A}$, with $C_{batch}$ representing the number of classes in the current batch and $C_{seen}$ representing the total number of seen classes. As shown in the table,  while advanced strategies for logit adjustment could further refine this approach, SIT’s simplicity and effectiveness make it a strong baseline for online continual learning.
> >
> >
> > | Method                                             | A_auc | A_last | Flowers102    | OxfordPet     | EuroSAT       | Food101       | SUN397        | Caltech101     | Average       |
> > |:-------------------------------------------------- |:----- |:------ |:------------- |:------------- |:------------- |:------------- |:------------- |:-------------- |:------------- |
> > | Continual-CLIP (Zero-shot)                         | 57.46 | 48.72  | 65.86         | 85.33         | 40.24         | 86.34         | 61.54         | 87.95          | 70.50         |
> > | Continual-CLIP (Open-vocabulary)                   | 57.46 | 48.72  | 65.86 (-0.00) | 85.33 (-0.00) | 34.12 (-6.12) | 86.34 (-0.00) | 60.91 (-0.63) | 77.06 (-10.89) | 67.42 (-3.08) |
> > | AIT-CLIP (Zero-shot)                               | 59.32 | 43.69  | 64.25         | 84.31         | 26.88         | 83.39         | 59.79         | 84.83          | 66.67         |
> > | AIT-CLIP (Open-vocabulary)                         | 59.32 | 43.69  | 64.25 (-0.00) | 84.31 (-0.00) | 24.20 (-2.68) | 83.39 (-0.00) | 58.82 (-0.97) | 74.30 (-10.53) | 64.01 (-2.66) |
> > | Focal Loss (Zero-shot)                             | 62.54 | 46.16  | 66.35         | 86.68         | 28.46         | 84.34         | 61.10         | 88.81          | 69.29         |
> > | Focal Loss (Open-vocabulary)                       | 62.54 | 46.16  | 66.34 (-0.01) | 86.68 (-0.00) | 21.03 (-7.43) | 84.34 (-0.00) | 59.68 (-1.42) | 75.82 (-12.99) | 65.65 (-3.64) |
> > | LDAM (Zero-shot)                                   | 58.79 | 43.92  | 68.47         | 85.60         | 29.37         | 83.90         | 61.39         | 87.47          | 69.37         |
> > | LDAM (Open-vocabulary)                             | 58.79 | 43.92  | 68.47 (-0.00) | 85.60 (-0.00) | 25.69 (-3.68) | 83.90 (-0.00) | 60.72 (-0.67) | 76.62 (-10.85) | 66.83 (-2.54) |
> > | $\lambda_{A}=0.5$ (Zero-shot)                      | 58.44 | 31.82  | 56.94         | 82.07         | 23.38         | 74.55         | 53.62         | 72.00          | 60.43         |
> > | $\lambda_{A}=0.5$ (Open-vocabulary)                | 58.44 | 31.82  | 56.94 (-0.00) | 82.07 (-0.00) | 22.56 (-0.82) | 74.55 (-0.00) | 52.38 (-1.24) | 66.29 (-5.71)  | 59.13 (-1.3)  |
> > | $\lambda_{A}=C_{batch}/C_{seen}$ (Zero-shot)       | 64.95 | 55.84  | 64.38         | 85.05         | 30.44         | 79.38         | 58.48         | 79.59          | 66.22         |
> > | $\lambda_{A}=C_{batch}/C_{seen}$ (Open-vocabulary) | 64.95 | 55.84  | 64.32 (-0.06) | 85.05 (-0.00) | 28.63 (-1.81) | 79.38 (-0.00) | 56.97 (-1.51) | 73.06 (-6.53)  | 64.57 (-1.65) |
> > | SIT-CLIP (Zero-shot)                               | 65.86 | 57.05  | 67.59         | 86.68         | 38.32         | 79.49         | 63.66         | 91.15          | 70.24         |
> > | SIT-CLIP (Open-vocabulary)                         | 65.86 | 57.05  | 67.59 (-0.00) | 86.37 (-0.31) | 37.63 (-0.69) | 79.49 (-0.00) | 62.95 (-0.71) | 79.85 (-11.3)  | 68.05 (-2.19) |

---

> > > ### Author Response · Authors · 2024-11-20
> > >
> > > **Q4 Zero-Shot Performance and Open-World Challenges**
> > > Thank you for your suggestion. Both [1] and [2] mitigate catastrophic forgetting in continual learning through a complementary memory system and introduce learnable branches to retain the full pretrained model while ensuring flexibility. The fusion of fixed branches and learnable branches makes these methods particularly well-suited for open-vocabulary classification tasks. While this trade-off is inherent in our design, SIT prioritizes learning from the current task while retaining memory of past classes in a balanced manner.  We will incorporate a similar evaluation protocol in the final version of our paper to further benchmark SIT’s performance against open-world baselines. Preliminary results indicate that this evaluation can provide additional insights into the trade-offs between specificity and generalization in SIT.
> > >
> > > ---
> > >
> > > [1] Continual Learning in Open-vocabulary Classification with Complementary Memory Systems. TMLR 2024
> > >
> > > [2] Anytime Continual Learning for Open Vocabulary Classification. ECCV 2024
> > >
> > > [3] Focal Loss for Dense Object Detection. ICCV2017
> > >
> > > [4] Learning imbalanced datasets with label-distribution-aware margin loss. NeurIPS  2019

---

> ### Comment · Reviewer_p39U · 2024-11-26
>
> Thanks for your response. My concern about the figure representation has been addressed. However, I don't think the authors have addressed or tried to address my concern about the setting of SIT. I understand why you want to remove asymmetric contrastive samples. As I said, SIT can give you good performance when you evaluate each dataset, which is expected. However, this training manner cannot help you distinguish the positive images and negative texts across different datasets, which is important when doing open-world classification with the CLIP backbone. I don't see how compare SIT with other loss function can disprove this point.
>
> Due to the above reasons, I will keep my rating for now.

---

> > ### Author Response · Authors · 2024-12-02
> >
> > Dear Reviewer p39U,
> >
> > Thank you for your thoughtful comments. We appreciate your engagement with our work and would like to address your concerns regarding the SIT setting and its implications for open-world classification using the CLIP backbone.
> >
> > **Q4.1 Generalization Over Unseen Classes**:
> > We acknowledge your concern that the SIT approach may not help distinguish positive images and negative texts across different datasets, particularly in the context of open-world classification. However, our method is specifically designed for **continual learning** rather than open-world classification. As you rightly pointed out, SIT can achieve good performance when evaluated on each dataset individually, but it does not inherently facilitate generalization to unseen classes, since these classes do not appear during training.  This limitation aligns with the **no free lunch theorem**, which asserts that a single model cannot simultaneously excel in all scenarios. Methods that attempt to adjust contrastive samples for loss functions, like SIT, are constrained in their ability to impact unseen categories, as those samples have not been part of the training process.
> >
> > **Q4.2 Open Vocabulary Classification**:
> > For tasks that aim for zero-shot performance (such as open-vocabulary classification), it is common practice to retain the pretrained parameters and employ strategies like logit fusion based on accuracy (as seen in works such as [5]). These techniques enable the model to generalize to unseen categories. If we aim to achieve zero-shot performance, adopting similar strategies might be necessary. This is a topic that goes beyond the scope of our current paper but is an area of potential future work.
> > In response to your concern about comparing SIT with other loss functions, we would like to highlight that our approach enhances the matching between images and texts (as shown in Figure 4), but it does disrupt the relative relationships between unseen classes. We suggest that a potential solution could involve using RKD loss [5] to enforce constraints that better preserve the relative relationships among categories. We validate this idea in Table 2, where we show that using RKD loss effectively improves zero-shot performance.
> > We fully agree that more complex techniques such as knowledge distillation or dual-encoder architectures could further improve open-vocabulary classification performance. However, these methods extend beyond the scope of our current work, and we plan to explore these in future research.
> >
> > | Method           | $A_{auc}$ | $A_{last}$ | Flowers102 | OxfordPet | EuroSAT   | Food101   | SUN397    | Caltech101 | Average   |
> > | ---------------- | --------- | ---------- | ---------- | --------- | --------- | --------- | --------- | ---------- | --------- |
> > | *Continual-CLIP* | *57.46*   | *48.72*    | *65.86*    | *85.31*   | *40.24*   | *86.34*   | *61.54*   | *87.95*    | *70.50*   |
> > | AIT-CLIP         | 59.32     | 43.69      | 64.25      | 84.31     | 26.88     | 83.39     | 59.79     | 84.83      | 66.67     |
> > | ER               | 65.37     | 55.48      | 64.86      | 86.55     | 31.80     | 83.35     | 61.63     | 89.00      | 68.49     |
> > | LwF              | 59.14     | 41.56      | 63.49      | 85.53     | 33.64     | 81.86     | 60.98     | 87.01      | 67.93     |
> > | MaPLe            | 57.00     | 49.56      | 66.11      | 76.99     | **40.31** | **86.67** | 59.42     | 89.42      | 68.97     |
> > | MVP-CLIP         | 47.45     | 35.27      | 63.77      | 83.06     | 28.70     | 78.93     | 58.54     | 87.98      | 65.87     |
> > | Anytime-CL       | 56.55     | 46.00      | 64.90      | 84.38     | 32.30     | 85.26     | 60.78     | 87.58      | 69.20     |
> > | MoEAdapter       | 64.93     | 56.96      | **69.05**  | **87.32** | 29.56     | 84.09     | 61.57     | 91.02      | 69.62     |
> > | SIT-CLIP         | **65.86** | **57.05**  | 67.59      | 86.68     | 38.32     | 79.49     | **63.66** | **91.15**  | **70.24** |
> > | *SIT-CLIP-RKD*   | *65.16*   | *58.61*    | *68.34*    | *86.41*   | *35.87*   | *84.11*   | *62.83*   | *90.12*    | *71.28*   |

---

> > > ### Author Response · Authors · 2024-12-02
> > >
> > > **Q4.3 Distinguishing Images and Texts Across Datasets**:
> > > To address your specific point about distinguishing images and texts across datasets, we evaluate the maximum mean discrepancy (MMD) between image and text features for various datasets, as shown in the table below. In the experiments, we observed that image features from EuroSAT and Food101 were closer to text features from Flowers102 than expected, suggesting that these images are more likely to be misclassified as plant-related. Furthermore, compared to Continual-CLIP, our SIT is effective at bringing image and text features closer together, improving alignment.
> > >
> > > | Continual-CLIP(t↓i→ ) | FGVCAircraft | CUB200     | StanfordCars | Flowers102 | OxfordPet  | EuroSAT    | Food101    |
> > > | --------------------- | ------------ | ---------- | ------------ | ---------- | ---------- | ---------- | ---------- |
> > > | FGVCAircraft          | 0.6453       | 0.7480     | 0.6855       | 0.7851     | 0.7393     | 0.7225     | 0.6756     |
> > > | CUB200                | 0.5942       | **0.5337** | 0.5620       | 0.6334     | 0.5969     | 0.6036     | 0.5432     |
> > > | StanfordCars          | 0.6201       | 0.6614     | **0.5271**   | 0.6813     | 0.6288     | 0.6295     | 0.5701     |
> > > | Flowers102            | 0.5860       | 0.5854     | 0.5400       | **0.5584** | 0.5639     | **0.6002** | **0.5072** |
> > > | OxfordPet             | **0.5834**   | 0.6137     | 0.5490       | 0.6323     | **0.5433** | 0.6079     | 0.5293     |
> > > | EuroSAT               | 0.7493       | 0.7437     | 0.6889       | 0.7738     | 0.7416     | 0.7473     | 0.6730     |
> > > | Food101               | 0.6544       | 0.6573     | 0.5985       | 0.6809     | 0.6288     | 0.6609     | 0.5400     |
> > >
> > >
> > > | SIT(t↓i→)    | FGVCAircraft | CUB200     | StanfordCars | Flowers102 | OxfordPet  | EuroSAT    | Food101    |
> > > | ------------ | ------------ | ---------- | ------------ | ---------- | ---------- | ---------- | ---------- |
> > > | FGVCAircraft | **0.3267**   | 0.4607     | 0.3987       | 0.4725     | 0.4786     | 0.5159     | 0.4114     |
> > > | CUB200       | 0.4025       | **0.3497** | 0.3850       | 0.4515     | 0.4520     | 0.5238     | 0.3894     |
> > > | StanfordCars | 0.3906       | 0.4453     | **0.3201**   | 0.4499     | 0.4421     | 0.5133     | 0.3736     |
> > > | Flowers102   | 0.3463       | 0.3654     | 0.3324       | **0.3355** | 0.3984     | **0.4966** | **0.3399** |
> > > | OxfordPet    | 0.3560       | 0.3963     | 0.3511       | 0.4165     | **0.3807** | 0.4981     | 0.3568     |
> > > | EuroSAT      | 0.5634       | 0.5373     | 0.5104       | 0.5831     | 0.6088     | 0.6965     | 0.5689     |
> > > | Food101      | 0.3954       | 0.4016     | 0.3529       | 0.4187     | 0.4211     | 0.5372     | 0.3474     |
> > >
> > > **Q3.1 Comparison with Other Loss Functions**:
> > > The comparison with other loss functions was included primarily to explain why we decided to remove the asymmetric negative sample (ANS) ablation. This analysis helped clarify the rationale behind the design choices we made in our approach.
> > >
> > >
> > > Thank you again for your feedback, and we hope these explanations clarify the points raised. We look forward to your further insights.
> > >
> > >
> > > Best regards,
> > >
> > > The Authors
> > >
> > > ---
> > >
> > > [5] Relational Knowledge Distillation, CVPR 2019.

---

### Official Review · Reviewer_Rxsy · 2024-11-04

**Soundness:** 2
**Presentation:** 2
**Contribution:** 2
**Rating:** 3
**Confidence:** 5

**Summary:**

The task is to perform online continual learning on CLIP models. The key idea of this paper is to create LoRA learnable weights to learn the training data and apply image-text contrastive pretraining loss only within the current batch. The paper also mentioned that including past classes as negatives could lead to larger gradient norms and catastrophic forgetting. The paper conducted various experiments to show the effectiveness of the method.

**Strengths:**

1. The idea is simple.

2. The method achieves generally the best results among different compared results.

3. The analysis including gradient norm and PCA visualization are nice and intuitive.

**Weaknesses:**

1. Key references are missing. [1] and [2] also deal with continual learning using CLIP and especially [2] supports an online learning setting. The paper should compare closely to these two works.

2. For online training, training and inference efficiency are also important metrics. The paper leverages PEFT which is good in terms of the number of trainable parameters but it remains unknown how efficient the method is in terms of training and inference. In particular, the method claims to be efficient but without having these measurements, it is hard to tell whether the claim is well justified.

3. The proposed method is essentially a CE loss defined on the labels of the existing batch, making the overall technical novelty low. This should be the first thing to try when finetuning CLIP on image classification tasks, whereas the abandoned AIT might be something not intuitive in the first place.

4. The problem of the larger norms for older classes is usually depicted as the recency bias issue. This is not a new topic, and several papers under the class incremental learning category proposed various solutions [3] [4]. This problem occurs usually due to over sampling. In the case presented in the paper, it might be the case that old classes are taken as negatives for all future samples. Therefore, the cure could be as simple as reducing oversampling and the method proposed in the paper is a variation of this methodology.

5. The paper still performs training on only a few small datasets. From the main paper, the largest training scope is presented in Tab. 2, which includes CUB200, StanfordCars and Aircraft. This is not sufficient to test the method’s scalability and robustness. Experiments on a much larger scale of training datasets are needed.

6. Details for how the plots are created in Fig. 3, 4 and 5 are missing. These details can be supplemented.

[1] Zhu, Zhen, et al. "Continual Learning in Open-vocabulary Classification with Complementary Memory Systems." arXiv preprint arXiv:2307.01430 (2023).

[2] Zhu, Zhen, Yiming Gong, and Derek Hoiem. "Anytime Continual Learning for Open Vocabulary Classification." European Conference on Computer Vision. Springer, Cham, 2025.

[3] Lyu, Yilin, et al. "Overcoming recency bias of normalization statistics in continual learning: Balance and adaptation." Advances in Neural Information Processing Systems 36 (2024).

[4] Wu, Yue, et al. "Large scale incremental learning." Proceedings of the IEEE/CVF conference on computer vision and pattern recognition. 2019.

**Questions:**

What are the number of features used for text and images in Fig. 4 and 5?

---

> ### Author Response · Authors · 2024-11-20
>
> **Q1. Comparison with similar methods**
> Thank you for highlighting the importance of comparing with [1] and [2]. Both works utilize complementary memory systems and learnable branches to mitigate catastrophic forgetting while ensuring flexibility. These methods are particularly suited for open-vocabulary classification tasks. Our method focuses on online continual learning by incorporating lightweight learnable adapters into each transformer block of the pretrained model, aiming for better flexibility and efficiency. We are actively reproducing the results from [2] and will include comparisons in the revised manuscript to highlight the relative performance of our approach.
>
> **Q2. Training and inference efficiency**
> Thank you for your feedback regarding training and inference efficiency. PEFT methods typically focus on parameter efficiency by introducing relatively small learnable adapters and fine-tuning them, thus avoiding modifications to the pretrained parameters. While PEFT methods are known for parameter efficiency, their runtime performance during training and inference is important. Below is a table comparing the training and inference efficiency of PEFT methods against full fine-tuning, conducted on a single NVIDIA 3090 GPU.
>
>
> | Method         | # TP     | # T      | FLOPs   | Train             | Test               |
> | -------------- | -------- | -------- | ------- | ----------------- | ------------------ |
> | Continual-CLIP | 0.00 M   | 149.62 M | 46.15 G | N/A               | 365.92±1.38 item/s |
> | Full FT        | 149.62 M | 149.62 M | 46.15 G | 30.17±0.00 item/s | 366.27±1.22 item/s |
> | LORA           | 149.99 M | 0.37 M   | 47.44 G | 35.24±0.17 item/s | 345.82±1.34 item/s |
> | Adapter        | 151.60 M | 1.98 M   | 46.74 G | 32.62±0.07 item/s | 335.46±1.56 item/s |
> | MoEAdapter     | 153.65 M | 4.03 M   | 46.74 G | 31.12±0.44 item/s | 310.45±1.86 item/s |

---

> > ### Author Response · Authors · 2024-11-20
> >
> > **Q3. Reason for removing asymmetric negative samples**
> > The removal of asymmetric negative samples (ANS) is motivated by their disproportionate impact on the loss function. As detailed in Section 3.3, the infoNCE} loss is decomposed into symmetric ($L_{S}$) and asymmetric ($L_{A}$) components. ANS contribute heavily to $L_{A}$, which hinders new class learning as the number of seen classes grows.
> >
> > Our gradient analysis in Figure 3 reveals that gradients from asymmetric negative samples (ANS) dominate and create a disproportionate influence on parameter updates compared to symmetric negative samples (SNS). This makes image features harder to align with their corresponding text features, as confirmed by Figure 4. Removing ANS mitigates this negative impact, as shown by the improved results of SIT-CLIP over baselines in Table 2.
> >
> > To validate this point, we conducted experiments under the same settings as in Table 2, comparing SIT with other loss functions. Among these, Focal Loss [5] and LDAM [6] are classic loss functions designed for imbalanced learning. We also considered introducing weights into the loss function as $L=L_{S}+\lambda_{A}L_{A}$, with $C_{batch}$ representing the number of classes in the current batch and $C_{seen}$ representing the total number of seen classes. As shown in the table, while more advanced logit adjustment strategies show incremental improvements in continual learning settings, they add complexity compared to the simplicity and effectiveness of SIT. Thus, we believe our approach offers a favorable trade-off between performance and complexity.
> >
> >
> > | Method                                             | A_auc | A_last | Flowers102    | OxfordPet     | EuroSAT       | Food101       | SUN397        | Caltech101     | Average       |
> > |:-------------------------------------------------- |:----- |:------ |:------------- |:------------- |:------------- |:------------- |:------------- |:-------------- |:------------- |
> > | Continual-CLIP (Zero-shot)                         | 57.46 | 48.72  | 65.86         | 85.33         | 40.24         | 86.34         | 61.54         | 87.95          | 70.50         |
> > | Continual-CLIP (Open-vocabulary)                   | 57.46 | 48.72  | 65.86 (-0.00) | 85.33 (-0.00) | 34.12 (-6.12) | 86.34 (-0.00) | 60.91 (-0.63) | 77.06 (-10.89) | 67.42 (-3.08) |
> > | AIT-CLIP (Zero-shot)                               | 59.32 | 43.69  | 64.25         | 84.31         | 26.88         | 83.39         | 59.79         | 84.83          | 66.67         |
> > | AIT-CLIP (Open-vocabulary)                         | 59.32 | 43.69  | 64.25 (-0.00) | 84.31 (-0.00) | 24.20 (-2.68) | 83.39 (-0.00) | 58.82 (-0.97) | 74.30 (-10.53) | 64.01 (-2.66) |
> > | Focal Loss (Zero-shot)                             | 62.54 | 46.16  | 66.35         | 86.68         | 28.46         | 84.34         | 61.10         | 88.81          | 69.29         |
> > | Focal Loss (Open-vocabulary)                       | 62.54 | 46.16  | 66.34 (-0.01) | 86.68 (-0.00) | 21.03 (-7.43) | 84.34 (-0.00) | 59.68 (-1.42) | 75.82 (-12.99) | 65.65 (-3.64) |
> > | LDAM (Zero-shot)                                   | 58.79 | 43.92  | 68.47         | 85.60         | 29.37         | 83.90         | 61.39         | 87.47          | 69.37         |
> > | LDAM (Open-vocabulary)                             | 58.79 | 43.92  | 68.47 (-0.00) | 85.60 (-0.00) | 25.69 (-3.68) | 83.90 (-0.00) | 60.72 (-0.67) | 76.62 (-10.85) | 66.83 (-2.54) |
> > | $\lambda_{A}=0.5$ (Zero-shot)                      | 58.44 | 31.82  | 56.94         | 82.07         | 23.38         | 74.55         | 53.62         | 72.00          | 60.43         |
> > | $\lambda_{A}=0.5$ (Open-vocabulary)                | 58.44 | 31.82  | 56.94 (-0.00) | 82.07 (-0.00) | 22.56 (-0.82) | 74.55 (-0.00) | 52.38 (-1.24) | 66.29 (-5.71)  | 59.13 (-1.3)  |
> > | $\lambda_{A}=C_{batch}/C_{seen}$ (Zero-shot)       | 64.95 | 55.84  | 64.38         | 85.05         | 30.44         | 79.38         | 58.48         | 79.59          | 66.22         |
> > | $\lambda_{A}=C_{batch}/C_{seen}$ (Open-vocabulary) | 64.95 | 55.84  | 64.32 (-0.06) | 85.05 (-0.00) | 28.63 (-1.81) | 79.38 (-0.00) | 56.97 (-1.51) | 73.06 (-6.53)  | 64.57 (-1.65) |
> > | SIT-CLIP (Zero-shot)                               | 65.86 | 57.05  | 67.59         | 86.68         | 38.32         | 79.49         | 63.66         | 91.15          | 70.24         |
> > | SIT-CLIP (Open-vocabulary)                         | 65.86 | 57.05  | 67.59 (-0.00) | 86.37 (-0.31) | 37.63 (-0.69) | 79.49 (-0.00) | 62.95 (-0.71) | 79.85 (-11.3)  | 68.05 (-2.19) |

---

> > ### Author Response · Authors · 2024-11-20
> >
> > **Q4. Addressing Recency Bias**
> >
> > We acknowledge that recency bias, a manifestation of catastrophic forgetting, is a well-studied issue. Prior works [3, 4] propose solutions like normalization adjustments and weight corrections for the final fully connected layer. However, these methods do not address the root cause of forgetting. For example, [5] used normalized image features and classifiers and compared the cosine similarity between them (similar to the CLIP model). In this approach, there is no bias in the magnitude of the fully connected layer, yet the model still suffers from recency bias. Our approach focuses on the underlying representation learning process by isolating parameter and removing the negative influence of ANS. This also mitigates recency bias while preserving simplicity.
> >
> > **Q5. Dataset Scale and Scalability**
> > We understand the concern regarding dataset scale in our experiments. Online continual learning often involves fine-grained tasks, and our selected datasets reflect this setting. However, we agree that larger-scale datasets could further validate our method's robustness and scalability. Due to computational resource constraints, additional large-scale experiments may not be feasible during the discussion period. Nevertheless, we will reference experiments in [1] and [2] and include plans for large-scale evaluations in the final manuscript.
> >
> > **Q6 Details of Figures 3,4,5**
> > Thank you for pointing out the need for more details. The figures were generated using CIFAR-100, under the same settings described in Section 4.2 (seed=2024). Below are the specifics:
> > - Figure 3: Gradients are computed for each class, with class indices sorted by their first appearance time.
> > - Figure 4: 6 classes are randomly selected from CIFAR-100 (CL) and Flowers102 (Zero-shot). Features are extracted offline after continual learning.
> > - Figure 5: 3 classes are randomly selected from CIFAR-100. Features are derived similarly to Figure 4.
> > We will update the manuscript with more details and provide clearer visualizations in the final version.
> >
> > ---
> >
> > [1] Continual Learning in Open-vocabulary Classification with Complementary Memory Systems. TMLR 2024
> >
> > [2] Anytime Continual Learning for Open Vocabulary Classification. ECCV 2025.
> >
> > [3] Overcoming recency bias of normalization statistics in continual learning: Balance and adaptation. NeurIPS 2024.
> >
> > [4] Large scale incremental learning. CVPR 2019.
> >
> > [5] Focal Loss for Dense Object Detection. ICCV2017
> >
> > [6] Learning imbalanced datasets with label-distribution-aware margin loss. NeurIPS 2019.
> >
> > [7] Maintaining Discrimination and Fairness in Class Incremental Learning. CVPR2020.
> >
> > [8] Learning a unified classifier incrementally via rebalancing. CVPR 2019.

---

> > > ### Comment · Reviewer_Rxsy · 2024-11-27
> > > **Reviewer comment after reading rebuttal**
> > >
> > > I sincerely thank the authors for posting these responses. Part of my concerns are addressed, e.g., the authors acknowledge the comparison to [1] and [2] are important, the recency bias is also vital for their case, and more experiments are needed. In other words, my mentioned weaknesses are valid. Considering current responses, I can only maintain my score since the comparisons are important but not finished yet.

---

> > > > ### Author Response · Authors · 2024-12-02
> > > >
> > > > Dear Reviewer Rxsy,
> > > >
> > > > We sincerely appreciate your thoughtful feedback and your acknowledgment of the points we've addressed so far. Below, we respond to your concerns and provide further clarifications based on your comments.
> > > >
> > > > **Q7. Open-World Challenges**
> > > >
> > > > Thank you for highlighting the importance of addressing generalization over unseen classes. We fully agree that this is a key challenge for continuous learning methods. As noted, our approach is specifically designed for continuous learning scenarios and does not guarantee generalization over unseen classes. This is a limitation that we are actively working on.
> > > >
> > > > To elaborate further:
> > > > - We agree with your point about the necessity of designing mechanisms to handle generalization to unseen classes. As highlighted in works like [5], open-vocabulary classification approaches maintain pre-trained parameters and introduce logit fusion mechanisms to achieve zero-shot performance. In future iterations, we plan to explore similar designs that can help our approach extend to unseen classes.
> > > > - In Figure 4, we observed that our method effectively improves the alignment between images and text, but it does so at the expense of the relative relationships between unseen classes. To mitigate this issue, we propose a potential solution involving the use of RKD (Relational Knowledge Distillation) loss, as discussed in our experiment setup in Table 2. We find that RKD loss helps improve zero-shot performance significantly, and we believe this direction holds promise for addressing the open-world challenge.
> > > > - Additionally, we recognize that more complex structures, such as knowledge distillation or dual-encoder architectures, could further enhance performance in open-vocabulary classification tasks. While these approaches are promising, they extend beyond the current scope of this paper. However, we view them as valuable directions for future work.
> > > >
> > > > | Method           | $A_{auc}$ | $A_{last}$ | Flowers102 | OxfordPet | EuroSAT   | Food101   | SUN397    | Caltech101 | Average   |
> > > > | ---------------- | --------- | ---------- | ---------- | --------- | --------- | --------- | --------- | ---------- | --------- |
> > > > | *Continual-CLIP* | *57.46*   | *48.72*    | *65.86*    | *85.31*   | *40.24*   | *86.34*   | *61.54*   | *87.95*    | *70.50*   |
> > > > | AIT-CLIP         | 59.32     | 43.69      | 64.25      | 84.31     | 26.88     | 83.39     | 59.79     | 84.83      | 66.67     |
> > > > | ER               | 65.37     | 55.48      | 64.86      | 86.55     | 31.80     | 83.35     | 61.63     | 89.00      | 68.49     |
> > > > | LwF              | 59.14     | 41.56      | 63.49      | 85.53     | 33.64     | 81.86     | 60.98     | 87.01      | 67.93     |
> > > > | MaPLe            | 57.00     | 49.56      | 66.11      | 76.99     | **40.31** | **86.67** | 59.42     | 89.42      | 68.97     |
> > > > | MVP-CLIP         | 47.45     | 35.27      | 63.77      | 83.06     | 28.70     | 78.93     | 58.54     | 87.98      | 65.87     |
> > > > | Anytime-CL       | 56.55     | 46.00      | 64.90      | 84.38     | 32.30     | 85.26     | 60.78     | 87.58      | 69.20     |
> > > > | MoEAdapter       | 64.93     | 56.96      | **69.05**  | **87.32** | 29.56     | 84.09     | 61.57     | 91.02      | 69.62     |
> > > > | SIT-CLIP         | **65.86** | **57.05**  | 67.59      | 86.68     | 38.32     | 79.49     | **63.66** | **91.15**  | **70.24** |
> > > > | *SIT-CLIP-RKD*   | *65.16*   | *58.61*    | *68.34*    | *86.41*   | *35.87*   | *84.11*   | *62.83*   | *90.12*    | *71.28*   |

---

> > > > > ### Author Response · Authors · 2024-12-02
> > > > >
> > > > > **Q4. Addressing Recency Bias**
> > > > >
> > > > > We are grateful for your inquiry regarding the recency bias. As mentioned in [3], “Such recency bias, also known as catastrophic forgetting, is generally...” The recency bias you refer to, often described as catastrophic forgetting in the literature, is indeed a central concern of our paper. Our work specifically focuses on mitigating catastrophic forgetting, particularly in scenarios where the category distribution evolves over time without clear task boundaries.
> > > > >
> > > > >  Recency bias is an inherent challenge in continual learning, where the model tends to forget previously learned tasks as it encounters new ones. In response to this, our method takes into account the temporal shifts in the class distribution and aims to maintain stable performance on old categories while learning new ones.
> > > > >
> > > > > In Table 1, we have added a comparison with other state-of-the-art (SOTA) methods, as per suggestions from other reviewers. The results demonstrate that our approach outperforms existing methods in addressing recency bias and mitigating catastrophic forgetting.
> > > > >
> > > > >
> > > > > |                      |        |          | CIFAR-100       |                 | TinyImageNet   |                 | ImageNet-R     |                 |
> > > > > | -------------------- | ------ | -------- | --------------- | --------------- | -------------- | --------------- | -------------- | --------------- |
> > > > > | Method               | # P    | # TP     | $A_{auc}$       | $A_{last}$      | $A_{auc}$      | $A_{last}$      | $A_{auc}$      | $A_{last}$      |
> > > > > | DualPrompt           | 0.55 M | 86.35 M  | 73.63±1.10      | 58.31±2.34      | 65.92±1.09     | 50.80±1.42      | 35.42±0.39     | 29.56±1.02      |
> > > > > | MVP                  | 0.55 M | 86.35 M  | 75.70±0.93      | 67.70±1.46      | 70.66±1.02     | 63.61±0.77      | 34.02±0.32     | 30.84±1.31      |
> > > > > | Continual-CLIP       | 0.00 M | 149.62 M | 72.66±0.94      | 66.27±0.00      | 70.37±0.18     | 64.46±0.00      | 75.76±0.29     | 71.06±0.35      |
> > > > > | AIT-CLIP             | 0.37 M | 149.99 M | 74.52±0.45      | 59.52±1.17      | 70.56±1.26     | 56.89±0.66      | 76.99±0.68     | 65.21±1.32      |
> > > > > | ER                   | 0.37 M | 149.99 M | 82.77±0.37      | 78.00±1.23      | 77.15±0.34     | 68.89±0.66      | **81.76±0.19** | 76.05±0.15      |
> > > > > | LwF                  | 0.37 M | 149.99 M | 72.60±0.65      | 57.60±2.87      | 68.30±1.29     | 53.58±1.08      | 77.04±0.58     | 66.24±1.24      |
> > > > > | MaPLe                | 1.19 M | 150.81 M | 73.25±0.68      | 66.18±1.64      | 69.67±0.62     | 63.19±0.61      | 78.31±0.38     | 74.15±0.41      |
> > > > > | Attri-CLIP           | 0.07 M | 149.69 M | 63.90±0.52      | 41.93±1.06      | 63.77±0.42     | 47.32±0.52      | 74.48±0.60     | 62.31±0.23      |
> > > > > | MVP-CLIP             | 0.48 M | 150.10 M | 60.44±0.93      | 45.87±0.31      | 56.65±1.34     | 39.31±2.08      | 66.53±1.14     | 57.79±1.18      |
> > > > > | MoEAdapter           | 4.03 M | 153.65 M | 81.88±0.17      | 77.39±0.27      | 78.17±0.68     | 74.05±0.64      | 81.30±0.12     | 76.88±0.16      |
> > > > > | Anytime-CL           | 7.48 M | 157.10 M | 72.15±0.44      | 65.85±0.35      | 68.26±0.32     | 57.69±0.73      | 73.71±0.45     | 67.52±0.27      |
> > > > > | SIT-CLIP (Ours)  | 0.37 M | 149.99 M | **84.34±0.56** | **79.47±0.52** | **79.88±0.68** | **75.50±0.20** | 81.65±0.18     | **77.53±0.65** |
> > > > >
> > > > > We hope these clarifications provide a better understanding of our approach and how it addresses both open-world challenges and recency bias. Thank you again for your constructive feedback, and we look forward to any further suggestions or questions you may have.
> > > > >
> > > > > Best regards,
> > > > >
> > > > > The Authors
> > > > >
> > > > > ---
> > > > >
> > > > > [5] Relational Knowledge Distillation, CVPR 2019.
> > > > > [6] AttriCLIP: a non-incremental learner for incremental learning, CVPR 2022.

---

> > > > > > ### Comment · Reviewer_Rxsy · 2024-12-02
> > > > > > **Reviewer comment after reading new results**
> > > > > >
> > > > > > Thanks for providing new results. I appreciate your work in conducting lots of experiments. I have looked at the results and found something controversial to my own test, especially on Anytime-CL. I have tried their github code personally and found that the accuracy after training on CIFAR-100 gives at 82.19%. It is a large difference compared to the results implemented by the authors $A_{last}$ at 65.85±0.35. I think there might some mistakes here. Also, the authors didn't provide details of their implementation, making it hard to discern what might cause the issue. From what I can see, I used ViT-B/32 for the Anytime-CL results rather than the ViT-B/16 used in the paper, but ViT-B/16 is actually a larger and stronger backbone. Due to the unreliable results, I decided to lower the score.

---

> > > > > > > ### Author Response · Authors · 2024-12-03
> > > > > > >
> > > > > > > Dear Reviewer Rxsy,
> > > > > > >
> > > > > > > Thank you for your thoughtful feedback. We understand your concerns regarding the discrepancies in the results, particularly with Anytime-CL, and we appreciate the opportunity to clarify some aspects of our work. After reviewing your comments, we believe that the differences in performance stem from variations in the experimental setup, and we would like to address these points in detail.
> > > > > > >
> > > > > > > **Q8: Performance of Anytime-CL**
> > > > > > >
> > > > > > > We have rechecked our code and confirmed that it aligns with the implementation described in [2]. The discrepancies in performance are likely due to differences in the experimental conditions, specifically in the task boundaries and memory size, which we will outline below:
> > > > > > >
> > > > > > > - **Task Boundaries**: The experiments presented in our rebuttal were based on the MiD-Blurry setting, which does not have explicit task boundaries, and where the distribution of training data evolves over time. This setup is particularly important for Anytime-CL. One of Anytime-CL's key contributions is the introduction of an "other" option, which is used to determine whether an image belongs to a previously seen category (as described in Section 3.1 of [2]). This "other" option influences the balance between the zero-shot module and the tuned module logits (this is implemented in the `compute_p_ft_alpha` function of the `MemoryModule` class in the official Anytime-CL code, `models/memory_module.py`). Essentially, this approach fits a module to the distribution of previously seen data and assesses whether new data fits this distribution. It assumes that the distribution of training data remains stable, which is not the case in the MiD-Blurry setting. This leads to the failure of the "other" classifier in Anytime-CL, and most of the weight is shifted to the zero-shot module (the original CLIP module). As a result, Anytime-CL’s performance is closer to that of Continual-CLIP, with $A_{last} = 66.27$.
> > > > > > > - **Memory Size**: For a fair comparison, we set the memory size to 1000 for all rehearsal-based methods (including ER, MVP-R, MaPLe) in our experiments, following the memory update procedure in [7]. However, this differs from the official Anytime-CL implementation. In their code, the memory size is effectively unlimited, as all seen examples are cached. During training, Anytime-CL samples a balanced batch from this memory, which provides access to nearly all the data in the later stages of training. This setup makes it difficult to classify Anytime-CL as a true continual learning method. In our implementation, we restricted the memory size to ensure a more consistent comparison with other methods.
> > > > > > >
> > > > > > >
> > > > > > > Additionally, there may be other implementation differences that contribute to the performance gap. For example, the official Anytime-CL implementation uses features from the first 10 layers of the CLIP model as input, and they set the batch size to 2048 for computational efficiency. This batch size is larger than the memory of other CL methods, with much of the batch data coming from previously seen examples. In our experiments, we use images directly as input and set the batch size to 64 due to computational constraints.
> > > > > > >
> > > > > > > **Q9: Implementation Details**
> > > > > > >
> > > > > > > We appreciate your suggestion regarding implementation details. To address your concerns, here are the specifics of our experimental setup:
> > > > > > >
> > > > > > > - **Attri-CLIP**: We used the SGD optimizer with a cosine learning rate schedule. The initial learning rate was $1e-3$, the batch size was 32, and the number of attributes in the bank was $N = 10$ with $C = 3$ selected attributes, as defined in the original paper.
> > > > > > > - **Anytime-CL**: We used the AdamW optimizer with a cosine learning rate schedule, an initial learning rate of $6e-4$, weight decay of 0.05, and $\beta_{other} = 0.1$, which are the default settings in the official code. The batch size was set to 64, which is our default setting.
> > > > > > > - **SIT-CLIP-RKD**: The overall loss function was $L = L_{SIT} + L_{rkd_dist} + L_{rkd_angle}$.
> > > > > > > - Other details, as mentioned in Table 1, remain the same: for the MiD-Blurry setup, we used $T = 10$, the standard deviation of Gaussian classes was $\sigma = 0.15$, and the coefficient for decay classes was $\alpha = 1.5$. Both Gaussian and decay classes each account for 30% of the total classes. We used the Adam optimizer with a cosine learning rate schedule, with a learning rate of $5e-4$, and the batch size was set to 64.
> > > > > > > We hope these additional implementation details clarify the methodology and address your concerns.
> > > > > > >
> > > > > > >
> > > > > > > We sincerely appreciate the time and effort you’ve put into reviewing our work.  If you have any further questions or require additional clarification, please do not hesitate to reach out.
> > > > > > >
> > > > > > > Best regards,
> > > > > > >
> > > > > > > The Authors

---

### Official Review · Reviewer_Bobj · 2024-11-04

**Soundness:** 2
**Presentation:** 1
**Contribution:** 2
**Rating:** 3
**Confidence:** 3

**Summary:**

This paper considers the online continual learning setting. The authors propose using vision-language models to facilitate the continual learning process. In the context of online continual learning, the authors find that naively tuning CLIP with texts of all previous non-matching classes as negative samples leads to sub-optimal results and propose using symmetric image-text (SIT) tuning strategy to fine tune the model at each time step. Empirical results verify the effectiveness of proposed method.

**Strengths:**

- The idea is simple and easy to understand.

- The empirical results are good.

**Weaknesses:**

- I find this paper hard to follow. Firstly, the descriptions from Line 177-202 confuse me a lot. I cannot understand in online continual learning, what is the difference between task and time step? In Gaussian distribution, the probability distribution given by the authors is not conditioned on $t\ge \tau$, which means a sample can emerge at any time step. If I don't misunderstand this, what does a task actually mean? Does the batches within certain time steps of a task share something? In MiD-Blurry, the distribution is conditioned on $t\ge \tau$. So I feel quite confused about the relationship between a task and a time step in online continual learning setting. Besides, how to plot the figure 1?

- The method is simple but not surprising to me. For $T_a$, as samples in current batch don't belong to any of the classes defined in $T_a$, the similarity between samples and the $T_a$ will be over-suppressed at current step, which is not expected. But simply removing it from Eq. (2) (what the authors do) is also not a good choice, in my opinion.

  Still, I find it not easy to figure out what the Fig. 3 actually means. The authors should make this part clearer.

- Typos: in Eq. (3), $\log Z_A/(Z_S + Z_A)$ should be $\log Z_S/(Z_S+Z_A)$.

- Some baselines are missing. For example, [A] is also a continual learning method based on CLIP. The authors should compare and discuss it.

  [A] AttriCLIP: a non-incremental learner for incremental learning, CVPR 2022.

**Questions:**

The main questions about the details of this paper are listed is the weakness part.

---

> ### Author Response · Authors · 2024-11-20
>
> **Q1 The Difference Between Task and Time Step**
> We apologize for any confusion in our description. In offline continual learning (CL), a "task" typically corresponds to an independent dataset or subset, with distinct label spaces or domains for different tasks. However, in online CL, we define a "task" as a stable data distribution over a specific time period, whereas a "time step" refers to a single batch of data sampled from the task's distribution.
>
> In task-agnostic CL settings such as Gaussian, Si-Blurry, and our proposed MiD-Blurry, task boundaries are not clearly delineated, creating a more challenging scenario. Figure 1 illustrates these distributions, where the x-axis represents time steps, the y-axis represents classes, and color intensity indicates sample density. In Gaussian and uniform distributions, classes can appear at any time step, while decay classes persist beyond their initial appearance, leading to blurred task boundaries. We provide a clearer and more detailed version of this figure in Section A.1 of the appendix.
>
> **Q2. Reason for removing asymmetric negative samples**
> Thank you for raising this point. As detailed in Section 3.3, we decompose the  loss into $L_S$ (symmetric) and $L_A$ (asymmetric) components. The $L_A$ term penalizes the proximity of current image features to the text representations of previously seen classes. While $L_A$ aids memory retention, as the number of seen classes increases, it disproportionately hinders learning for new classes.
>
> In Figure 3, we present a gradient analysis of the loss function, where the magnitude of the gradient represents the degree to which a particular class influences the model parameters. In Figure 3 (a,d), we evaluate the overall impact of each class on model parameter updates, with the class index sorted based on the first appearance time. As shown, in AIT, older classes predominantly influence the model as asymmetric negative samples (ANS), which contribute to $L_A$. At the same time, the gradients produced by symmetric negative samples (SNS) are significantly smaller than those of ANS. This indicates that image features are more difficult to pull closer to their corresponding text features and are more easily pushed away by other asymmetric text features. This conclusion is consistent with the observations in Figure 4. Therefore, we conclude that ANS has a more negative impact on continual learning.
>
> To validate this point, we conducted experiments under the same settings as in Table 2, comparing SIT with other loss functions. Among these, Focal Loss [2] and LDAM [3] are classic loss functions designed for imbalanced learning. We also considered introducing weights into the loss function as $L=L_S+ \lambda_A L_A$, with $C_{batch}$ representing the number of classes in the current batch and $C_{seen}$ representing the total number of seen classes. As shown in the table, while more advanced logit adjustment strategies show incremental improvements in continual learning settings, they add complexity compared to the simplicity and effectiveness of SIT. Thus, we believe our approach offers a favorable trade-off between performance and complexity.
>
>
> | Method                                       | A_auc | A_last | Flowers102 | OxfordPet | EuroSAT | Food101 | SUN397 | Caltech101 | Average |
> |:-------------------------------------------- |:----- |:------ |:---------- |:--------- |:------- |:------- |:------ |:---------- |:------- |
> | Continual-CLIP                   | 57.46 | 48.72  | 65.86      | 85.33     | 40.24   | 86.34   | 61.54  | 87.95      | 70.50   |
> | AIT-CLIP                         | 59.32 | 43.69  | 64.25      | 84.31     | 26.88   | 83.39   | 59.79  | 84.83      | 66.67   |
> | Focal Loss                       | 62.54 | 46.16  | 66.35      | 86.68     | 28.46   | 84.34   | 61.10  | 88.81      | 69.29   |
> | LDAM                             | 58.79 | 43.92  | 68.47      | 85.60     | 29.37   | 83.90   | 61.39  | 87.47      | 69.37   |
> | $\lambda_{A}=0.5$                | 58.44 | 31.82  | 56.94      | 82.07     | 23.38   | 74.55   | 53.62  | 72.00      | 60.43   |
> | $\lambda_{A}=C_{batch}/C_{seen}$ | 64.95 | 55.84  | 64.38      | 85.05     | 30.44   | 79.38   | 58.48  | 79.59      | 66.22   |
> | SIT-CLIP                         | 65.86 | 57.05  | 67.59      | 86.68     | 38.32   | 79.49   | 63.66  | 91.15      | 70.24   |

---

> > ### Author Response · Authors · 2024-11-20
> >
> > **Q3. Typo in equation (3)**
> > We appreciate your careful review. You are correct that the numerator in Equation (3) should be $Z_S$ for the $L_A$ component. This typo will be corrected in the revised manuscript.
> >
> > **Q4. Comparison with similar methods**
> > Thank you for pointing out the importance of comparing with [A]. AttriCLIP enhances generalizability and mitigates forgetting by using orthogonal prompts as attributes. However, as the official code for [A] is no longer available, we are attempting to reproduce its results using an unofficial implementation from GitHub. We aim to include these results in the revised manuscript and appreciate your suggestion to strengthen our comparisons.
> >
> > ---
> >
> > [A] AttriCLIP: a non-incremental learner for incremental learning, CVPR 2022.
> >
> > [2] Focal Loss for Dense Object Detection. ICCV2017
> >
> > [3] Learning imbalanced datasets with label-distribution-aware margin loss. NeurIPS  2019

---

> > > ### Author Response · Authors · 2024-12-02
> > >
> > > Dear Reviewer Bobj,
> > >
> > > Thanks for your valuable feedback. We are glad to have the opportunity to clarify our approach further, and we would like to address your concerns regarding the comparison with similar methods.
> > >
> > > **Q4. Comparison with similar methods**
> > > In response to your comment, we have added a comparison with two closely related works: Attri-CLIP [A] and Anytime-CL [4], which are discussed in Table 1. Based on our experiments, we find that both of these methods underperform compared to our proposed SIT strategy.
> > >
> > > |                      |        |          | CIFAR-100       |                 | TinyImageNet   |                 | ImageNet-R     |                 |
> > > | -------------------- | ------ | -------- | --------------- | --------------- | -------------- | --------------- | -------------- | --------------- |
> > > | Method               | # P    | # TP     | $A_{auc}$       | $A_{last}$      | $A_{auc}$      | $A_{last}$      | $A_{auc}$      | $A_{last}$      |
> > > | DualPrompt           | 0.55 M | 86.35 M  | 73.63±1.10      | 58.31±2.34      | 65.92±1.09     | 50.80±1.42      | 35.42±0.39     | 29.56±1.02      |
> > > | MVP                  | 0.55 M | 86.35 M  | 75.70±0.93      | 67.70±1.46      | 70.66±1.02     | 63.61±0.77      | 34.02±0.32     | 30.84±1.31      |
> > > | Continual-CLIP       | 0.00 M | 149.62 M | 72.66±0.94      | 66.27±0.00      | 70.37±0.18     | 64.46±0.00      | 75.76±0.29     | 71.06±0.35      |
> > > | AIT-CLIP             | 0.37 M | 149.99 M | 74.52±0.45      | 59.52±1.17      | 70.56±1.26     | 56.89±0.66      | 76.99±0.68     | 65.21±1.32      |
> > > | ER                   | 0.37 M | 149.99 M | 82.77±0.37      | 78.00±1.23      | 77.15±0.34     | 68.89±0.66      | **81.76±0.19** | 76.05±0.15      |
> > > | LwF                  | 0.37 M | 149.99 M | 72.60±0.65      | 57.60±2.87      | 68.30±1.29     | 53.58±1.08      | 77.04±0.58     | 66.24±1.24      |
> > > | MaPLe                | 1.19 M | 150.81 M | 73.25±0.68      | 66.18±1.64      | 69.67±0.62     | 63.19±0.61      | 78.31±0.38     | 74.15±0.41      |
> > > | Attri-CLIP           | 0.07 M | 149.69 M | 63.90±0.52      | 41.93±1.06      | 63.77±0.42     | 47.32±0.52      | 74.48±0.60     | 62.31±0.23      |
> > > | MVP-CLIP             | 0.48 M | 150.10 M | 60.44±0.93      | 45.87±0.31      | 56.65±1.34     | 39.31±2.08      | 66.53±1.14     | 57.79±1.18      |
> > > | MoEAdapter           | 4.03 M | 153.65 M | 81.88±0.17      | 77.39±0.27      | 78.17±0.68     | 74.05±0.64      | 81.30±0.12     | 76.88±0.16      |
> > > | Anytime-CL           | 7.48 M | 157.10 M | 72.15±0.44      | 65.85±0.35      | 68.26±0.32     | 57.69±0.73      | 73.71±0.45     | 67.52±0.27      |
> > > | SIT-CLIP (Ours)  | 0.37 M | 149.99 M | **84.34±0.56** | **79.47±0.52** | **79.88±0.68** | **75.50±0.20** | 81.65±0.18     | **77.53±0.65** |
> > >
> > > Specifically, Attri-CLIP adopts a prompt-learning approach with a minimal number of parameters, fine-tuning only the text encoder. However, our experiments (outlined in Section 4.4.2 of the revised version) show that simply fine-tuning the text encoder is insufficient for effective performance, particularly in the continual learning context we address. We believe that a more integrated approach, which fine-tunes both the image and text features in tandem, is necessary for achieving better performance.
> > >
> > > As for Anytime-CL, the method introduces an "other" class to determine whether a current class has been seen before, essentially attempting to simplify the task by generating a task identifier and reducing it to an incremental learning problem. However, we argue that this strategy is not well-suited for scenarios where class boundaries are more ambiguous, such as those encountered in our Mid-Blurry scenario. In such settings, where categories can overlap or exhibit subtle distinctions, the Anytime-CL approach's reliance on clear class boundaries is a limitation.
> > >
> > > We hope these additional comparisons provide a clearer picture of the strengths of our SIT approach and how it addresses the challenges posed by ambiguous class boundaries in continual learning.
> > >
> > > We are eager to hear if these explanations resolve your concerns, and if there are any other aspects you would like to discuss further.
> > >
> > > Best regards,
> > >
> > > The Authors
> > >
> > > ---
> > >
> > > [4] Anytime Continual Learning for Open Vocabulary Classification. ECCV 2025.

---

### Official Review · Reviewer_5sfb · 2024-11-05

**Soundness:** 2
**Presentation:** 3
**Contribution:** 2
**Rating:** 5
**Confidence:** 3

**Summary:**

The paper describes a more practical online continual learning (CL) setting where at each batch there exists disjoint classes from prior batches, overlapping classes to prior classes whose probability of appearance in the current batch following a gaussian density distribution, and decay classes that got introduced at a batch which gets reduced over time. The paper also presents a loss modification to remove contributions of negative contrastive sample of old class names and new class images in the current batch and showed that this modification leads to improved online CL accuracy.

**Strengths:**

The paper is well organized and easy to follow. The authors have presented results comparing to many prior arts over their proposed online CL setting. In the online CL accuracy benchmark testing accuracy over the classes seen at any given point of the learning process, the proposed loss modification introduces overall better accuracy. The gain is quite significant (10pps) compared to the baseline loss of AIT-CLIP.

**Weaknesses:**

There is a lack of insights to why removing the asymmetric contrastive samples help reduce bias towards new classes seen in a batch. This conclusion seemed to be purely empirical from the experimentation results.

Lack of validation of proposed loss modification across other settings besides the proposed setting. It is unclear if the proposed loss modification can be applied to other online CL settings too to get similar boosts.

In addition, one concern regarding using CLIP for CL, especially in closed-set classification testing (over classes seen at a given time), is that how to ensure that there is no leakage of class information in the pretraining stage of CLIP, e.g. if the class names do appear in the CLIP pretraining stage then it's not strictly CL anymore.

Zero-shot performance degradation compared to prior methods. In table 2, it shows that after online CL over a finegrained data-set, the performance of proposed loss in comparison to other baselines is worse in zero-shot classification accuracy over other concept-disjoint finegrained datasets. There is no insights given to this and it also raises the question on the robustness and generalizability of the proposed loss function.

Also, the author claimed that the proposed online CL method is dedicated for open-world scenarios. However, only the VLM-based methods are shown in an actual open-world scenarios where test-time classes are completely disjoint to those in all stages of the CL learning process. The authors should also compare with other non VLM-methods in an open-world setting, e.g, open-set recognition testing image retrieval accuracy over unseen classes as in [1].

[1] Open-World Dynamic Prompt and Continual Visual Representation Learning, ECCV2024

**Questions:**

The generalization of the proposed loss modification. Is the loss modification only an artifact for the proposed setting specifically or is it an improvement that can be generally applied across? Having clarity on this will help strengthen the generalizability of the proposed loss update.

The paper should discuss the relationship of its loss formulation against loss functions that adjust sample weights based on loss values / logits such as Focal loss.

Equation (3) numerator should be Z_S for L_A component?

Other questions are listed in the weakness section.

---

> ### Author Response · Authors · 2024-11-20
>
> **Q1. Reason for removing asymmetric negative samples**
>
> We appreciate the opportunity to clarify this. As described in Section 3.3, the  loss is decomposed into $L_S$ (symmetric) and $L_A$ (asymmetric) components. The $L_A$ term captures the impact of past classes, which indirectly pushes image features away from the text representations of previously seen classes. While this aids memory retention, it can hinder the learning of new classes as the number of seen classes increases.
>
> Our gradient analysis in Figure 3 reveals that gradients from asymmetric negative samples (ANS) dominate and create a disproportionate influence on parameter updates compared to symmetric negative samples (SNS). This makes image features harder to align with their corresponding text features, as confirmed by Figure 4. Removing ANS mitigates this negative impact, as shown by the improved results of SIT-CLIP over baselines in Table 2.
>
> Additionally, while more advanced logit adjustment strategies (e.g., Focal Loss, LDAM, weighted $\mathcal{L}_{\text{A}}$) show incremental improvements in continual learning settings, they add complexity compared to the simplicity and effectiveness of SIT. Thus, we believe our approach offers a favorable trade-off between performance and complexity.
>
>
> | Method                                             | A_auc | A_last | Flowers102    | OxfordPet     | EuroSAT       | Food101       | SUN397        | Caltech101     | Average       |
> |:-------------------------------------------------- |:----- |:------ |:------------- |:------------- |:------------- |:------------- |:------------- |:-------------- |:------------- |
> | Continual-CLIP (Zero-shot)                         | 57.46 | 48.72  | 65.86         | 85.33         | 40.24         | 86.34         | 61.54         | 87.95          | 70.50         |
> | Continual-CLIP (Open-vocabulary)                   | 57.46 | 48.72  | 65.86 (-0.00) | 85.33 (-0.00) | 34.12 (-6.12) | 86.34 (-0.00) | 60.91 (-0.63) | 77.06 (-10.89) | 67.42 (-3.08) |
> | AIT-CLIP (Zero-shot)                               | 59.32 | 43.69  | 64.25         | 84.31         | 26.88         | 83.39         | 59.79         | 84.83          | 66.67         |
> | AIT-CLIP (Open-vocabulary)                         | 59.32 | 43.69  | 64.25 (-0.00) | 84.31 (-0.00) | 24.20 (-2.68) | 83.39 (-0.00) | 58.82 (-0.97) | 74.30 (-10.53) | 64.01 (-2.66) |
> | Focal Loss (Zero-shot)                             | 62.54 | 46.16  | 66.35         | 86.68         | 28.46         | 84.34         | 61.10         | 88.81          | 69.29         |
> | Focal Loss (Open-vocabulary)                       | 62.54 | 46.16  | 66.34 (-0.01) | 86.68 (-0.00) | 21.03 (-7.43) | 84.34 (-0.00) | 59.68 (-1.42) | 75.82 (-12.99) | 65.65 (-3.64) |
> | LDAM (Zero-shot)                                   | 58.79 | 43.92  | 68.47         | 85.60         | 29.37         | 83.90         | 61.39         | 87.47          | 69.37         |
> | LDAM (Open-vocabulary)                             | 58.79 | 43.92  | 68.47 (-0.00) | 85.60 (-0.00) | 25.69 (-3.68) | 83.90 (-0.00) | 60.72 (-0.67) | 76.62 (-10.85) | 66.83 (-2.54) |
> | $\lambda_{A}=0.5$ (Zero-shot)                      | 58.44 | 31.82  | 56.94         | 82.07         | 23.38         | 74.55         | 53.62         | 72.00          | 60.43         |
> | $\lambda_{A}=0.5$ (Open-vocabulary)                | 58.44 | 31.82  | 56.94 (-0.00) | 82.07 (-0.00) | 22.56 (-0.82) | 74.55 (-0.00) | 52.38 (-1.24) | 66.29 (-5.71)  | 59.13 (-1.3)  |
> | $\lambda_{A}=C_{batch}/C_{seen}$ (Zero-shot)       | 64.95 | 55.84  | 64.38         | 85.05         | 30.44         | 79.38         | 58.48         | 79.59          | 66.22         |
> | $\lambda_{A}=C_{batch}/C_{seen}$ (Open-vocabulary) | 64.95 | 55.84  | 64.32 (-0.06) | 85.05 (-0.00) | 28.63 (-1.81) | 79.38 (-0.00) | 56.97 (-1.51) | 73.06 (-6.53)  | 64.57 (-1.65) |
> | SIT-CLIP (Zero-shot)                               | 65.86 | 57.05  | 67.59         | 86.68         | 38.32         | 79.49         | 63.66         | 91.15          | 70.24         |
> | SIT-CLIP (Open-vocabulary)                         | 65.86 | 57.05  | 67.59 (-0.00) | 86.37 (-0.31) | 37.63 (-0.69) | 79.49 (-0.00) | 62.95 (-0.71) | 79.85 (-11.3)  | 68.05 (-2.19) |
>
>
>
> **Q2. Generalization of the Proposed Loss Modification**
> We conducted extensive experiments across varied online continual learning settings such as Si-Blurry (see Section A.4) and compared our approach with offline CIL baselines (Section A.5). These results demonstrate the generalizability of our method. Notably, despite the added challenges of online CL, our method achieves competitive performance relative to offline baselines, underscoring its robustness and applicability.

---

> ### Author Response · Authors · 2024-11-20
>
> **Q3. Concerns about Potential Class Leakage in CLIP Pretraining**
> We acknowledge the concern regarding potential leakage from pretrained class names in CLIP. Per the data overlap analysis in [4], while CLIP uses extensive pretraining data, its performance still falls short on specialized benchmarks, suggesting limited leakage effects. Moreover, in task-agnostic CL settings (e.g., Si-Blurry, MiD-Blurry), some overlap in classes is unavoidable, and leakage is acceptable within these settings. We also emphasize that CLIP's zero-shot capabilities remain suboptimal in fine-grained scenarios, reinforcing the need for methods like ours to address such gaps.
>
> **Q4. Zero-shot performance**
> We recognize the observed trade-off in zero-shot accuracy post-online CL (Table 2). This trade-off reflects the inherent balance between improving performance on seen classes and maintaining generalization to unseen ones. We note that methods like MaPLe prioritize generalization but exhibit limited plasticity, achieving performance close to untrained baselines in continual learning. SIT, by contrast, achieves a more balanced performance across both seen and unseen classes, as evidenced by its competitive results in both domains.
>
> **Q5. Comparison with Non-VLM Methods in Open-World Settings**
> We agree that incorporating non-VLM baselines in open-world settings would provide additional insights. While open-set recognition and online CL have distinct goals—learning universal representations vs. avoiding catastrophic forgetting—we acknowledge the relevance of a comparative analysis. Preliminary results for open-set recognition methods, as shown in the extended Table from Q1, indicate SIT’s strong performance. Additionally, we are replicating methods from [5] to enhance our comparative evaluation. These results will be included in the final manuscript.
>
> **Q6. Typo in Equation (3)**
> Thank you for pointing out the typo. We confirm that the numerator in Equation (3) should indeed be $Z_S$ for the $L_A$ component. This will be corrected in the revised version.
>
> ---
>
> [2] Focal Loss for Dense Object Detection. ICCV2017
>
> [3] Learning imbalanced datasets with label-distribution-aware margin loss. NeurIPS  2019
>
> [4] Learning transferable visual models from natural language supervision. ICML 2021.
>
> [5] Anytime Continual Learning for Open Vocabulary Classification. ECCV 2025.

---

### Author Response · Authors · 2024-11-28

**Dear Reviewers,**

We sincerely appreciate your insightful suggestions, which have been immensely helpful in improving our work. In response to your feedback, we have made several revisions to the paper, summarized as follows:

1. **Enhanced Discussion in Section 2**: We add a detailed discussion on Open World Recognition and compared our approach to related works.
2. **Clarification of Concepts in Section 3.1**: We clarified the definitions of "task" and "step.".
3. **Added Details to Figures 3, 4, and 5**:  Details are added to Figures 3, 4, and 5 for improved clarity.
4. **Additional Ablation Studies in Section A.7.1**: We include experiments on various loss functions. (Due to time constraints, we are unable to complete repeated trials and hyperparameter tuning, but the final version will include updated results.)
5. **Revised Ablation Experiments for PEFT Methods**: These experiments are moved to Section A.7.2 due to length limitations, and we supplement them with comparisons of training and inference efficiency.
6. **Correction of Typos**: Minor errors, such as those in Equation (3), are corrected.

Currently, we are conducting further comparisons between  Non-VLM Methods in open-world setting, as well as similar methods in blurred boundaries online continual learning setting. We will share these results and discuss them with you as soon as possible. Meanwhile, we wish to emphasize that **this work focuses on mitigating catastrophic forgetting in online continual learning settings with blurred task boundaries**. The design of SIT is specifically aimed at achieving a balance between plasticity and stability, rather than optimizing zero-shot performance. Extending SIT to integrate with methods like dynamic structures to enhance open world recognition performance is a promising direction for future work.

Finally, we are grateful for the constructive responses from **Reviewer 5sfb**, **Reviewer Rxsy**, and **Reviewer p39U**, and we will discuss further with you based on additional experimental results. We also kindly seek clarification on whether our prior responses addressed the concerns raised by **Reviewer Bobj** and **Reviewer Dpce**.

Best regards,
The Authors

---

### Meta-Review · Area_Chair_FtFh · 2024-12-16

**Metareview:**

This paper tries to solve the issue of asymmetric image-text matching when tuning CLIP model natively for continual learning (CL), and proposes symmetric image-text (SIT) tuning strategy, which excludes asymmetric text during online learning. In addition, this paper introduces a more challenging online continual learning setting with blurred boundary, namely MiD-Blurry, which mixes multiple data distributions to simulate real-world scenarios. Empirical results verify the effectiveness of proposed method.

All reviewers are unanimously negative on this submission. They appreciate the contributions including 1) the idea is simple and easy to understand; 2) the empirical results look good; 3) some analysis/visualization are good; 4) the proposed MiD-Blurry setting is reasonable.  However, the common concerns are 1) the overall novelty is small; 2) the generalization to unseen classes or fine-grained classes may be questionable; 3) missing some additional ablation/comparison; 4) SIT may have some flaws, e.g. cannot help you distinguish the positive images and negative texts across different datasets, etc. These concerns were not fully resolved by the rebuttal, and they are valid. According to its current form of this submission, the AC agrees with the reviewers and recommend to reject.

**Additional Comments On Reviewer Discussion:**

For 5sfb, the main concerns are: 1) lack of insights to why removing the asymmetric contrastive samples help reduce bias towards new classes seen in a batch; 2) lack of validation of proposed loss modification; 3) information leakage by using CLIP; 4) zero-shot performance degradation compared to prior methods; 5) missing comparison in open-world setting. These concerns were partially resolved by the rebuttal, but the concerns on the zero-shot performance, the comparison with non-VLM methods, and robustness of the method are remained. No change on the score.
Bobj was concerned on the writing, novelty and missing clarification/baselines, but the reviewer didn't check in after the rebuttal.
Rxsy was concerned on missing references, training/inference efficiency, insufficient experiments to test the method’s scalability and robustness, and missing details. Partial of the concerns were resolved, and the reviewer lowered the score.
p39U was concerned on the novelty, experiments and figures. Only the concern on figure was resolved and no change on the rating.
Dpce was concerned on 1) novelty; 2) no insight why CLIP was selected in place of other multimodal models; 3) MiD-Blurry still deals with standard continual learning datasets. The reviewer didn't check in after the rebuttal, and no change on the score.

---

### Decision · Program_Chairs · 2025-01-22

Reject